# Somite morphogenesis is required for axial blood vessel formation during zebrafish embryogenesis

Eric Paulissen[1], Nicholas J Palmisano[1], Joshua S Waxman[2,3], Benjamin L Martin[1]*

[1]Department of Biochemistry and Cell Biology, Stony Brook University, Stony Brook, United States; [2]Molecular Cardiovascular Biology Division and Heart Institute, Cincinnati Children's Hospital Medical Center, Cincinnati, United States; [3]Department of Pediatrics, University of Cincinnati College of Medicine, Cincinnati, United States

**Abstract** Angioblasts that form the major axial blood vessels of the dorsal aorta and cardinal vein migrate toward the embryonic midline from distant lateral positions. Little is known about what controls the precise timing of angioblast migration and their final destination at the midline. Using zebrafish, we found that midline angioblast migration requires neighboring tissue rearrangements generated by somite morphogenesis. The somitic shape changes cause the adjacent notochord to separate from the underlying endoderm, creating a ventral midline cavity that provides a physical space for the angioblasts to migrate into. The anterior to posterior progression of midline angioblast migration is facilitated by retinoic acid-induced anterior to posterior somite maturation and the subsequent progressive opening of the ventral midline cavity. Our work demonstrates a critical role for somite morphogenesis in organizing surrounding tissues to facilitate notochord positioning and angioblast migration, which is ultimately responsible for creating a functional cardiovascular system.

## Editor's evaluation

This article elegantly combines in vivo imaging and cell transplantation studies in zebrafish embryos to reveal how somite morphogenesis impacts early blood vessel formation. The study provides evidence that retinoic acid-driven somite remodeling generates a midline cavity that is critical to the recruitment and coalescence of angioblast blood vessel precursors at the midline. This work provides important new insights into how angioblast migration and blood vessel formation are coordinated by the timely rearrangement of surrounding tissues during embryonic development.

## Introduction

Early organismal development relies on a variety of tissues that collectively organize into a functional body plan. In the trunk of the vertebrate embryo, many tissues are differentiating at once and often interact with one another to influence their definitive structure (*McMillen and Holley, 2015*). The axial (notochord), paraxial (somites), and lateral (angioblasts and other tissues) mesoderm begins to be specified during gastrulation, and new cells are added to these tissues from posteriorly localized progenitor cells as they expand after gastrulation (*Kimelman, 2016*; *Martin, 2016*; *Martin and Kimelman, 2012*; *Row et al., 2016*). By the end of embryogenesis, the notochord is a prominent midline tissue of the trunk and provides structural support to the body and signals that pattern adjacent tissues (*Balmer et al., 2016*; *Glickman et al., 2003*; *Kimmel et al., 1995*). Pairs of somites flank either side of the notochord and give rise to tissues that include the skeletal muscle, tendons, and bone (*Tani et al., 2020*). Ventral to the notochord and between the somites, the axial vasculature

*For correspondence: benjamin.martin@stonybrook.edu

Competing interest: The authors declare that no competing interests exist.

structures of the dorsal aorta and posterior cardinal vein distribute blood throughout the embryo (*Hogan and Bautch, 2004*; *Isogai et al., 2001*).

During somitogenesis, newly formed somites undergo a maturation event in which they change their morphology from a cuboidal shape to a chevron shape and extend in the dorsal-ventral axis (*Kimmel et al., 1995*; *Tlili et al., 2019*). Many cellular rearrangements and mechanical stresses contribute to making the final definitive somite shape (*Hollway et al., 2007*; *Leal et al., 2014*; *Tlili et al., 2019*; *Yin and Solnica-Krezel, 2007*; *Yin et al., 2018*; *Youn and Malacinski, 1981*). At the same time that somites are changing shape, endothelial progenitors called angioblasts arise in the lateral plate mesoderm and migrate to the midline of the embryo to form the axial vasculature (*Jin et al., 2005*). Interestingly, angioblast migration and new somite formation occur in an anterior to posterior progression, with a wave of angioblast migration happening slightly after the wave of somitogenesis (*Jin et al., 2005*; *Kohli et al., 2013*; *Yabe and Takada, 2016*). Despite these two events occurring in close temporal and physical proximity to one another, it is not clear how they influence one another during this developmental stage. While some evidence has shown a relationship between somites and blood vessel patterning, much of this research focused on angiogenic sprouting or arterial-venous specification, well after angioblast migration and somitogenesis is completed (*Lawson et al., 2002*; *Shaw et al., 2006*; *Therapontos and Vargesson, 2010*; *Torres-Vázquez et al., 2004*).

Loss-of-function mutation of the notochord specifying gene *noto* results in nonautonomous migration defects of angioblasts as they move to the midline (*Fouquet et al., 1997*; *Helker et al., 2015*). In the absence of *noto,* the notochord adopts a somitic fate (*Fouquet et al., 1997*; *Talbot et al., 1995*). The requirement of notochord specification for angioblast migration suggested that the notochord may act to attract angioblasts to the midline. This model was further explored when a notochord-derived secreted factor named *apela* (also known as *toddler/elabela*) was discovered (*Chng et al., 2013*; *Freyer et al., 2017*; *Helker et al., 2015*; *Pauli et al., 2014*). This small peptide is expressed in the notochord and when mutated caused angioblast migration defects. However, *apela* loss of function also causes defects in mesoderm and endoderm formation prior to notochord and angioblast specification via gastrulation defects (*Freyer et al., 2017*; *Norris et al., 2017*; *Pauli et al., 2014*). Similarly, *noto* loss of function causes morphological changes that could interfere with angioblast migration, including a broad expansion of somite tissue near the developing blood vessels (*Halpern et al., 1995*). Thus, although it appears that mutations that affect notochord development can disrupt angioblast migration, the exact mechanism of this effect is not clear.

As new somites form during body axis extension, they secrete all-trans retinoic acid (RA). RA is a metabolic derivative of vitamin A, and a series of alcohol and aldehyde dehydrogenases convert vitamin A into RA in specific locations of the embryo including newly formed somites (*Duester, 2008*). RA acts as a morphogen with broad roles during vertebrate embryogenesis. Some of these include the development of the limb, skeleton, hindbrain, and heart (*Emoto et al., 2005*; *Heine et al., 1986*; *Lohnes et al., 1994*; *Mendelsohn et al., 1994*; *Niederreither et al., 2001*; *Sandell et al., 2007*). Although involved in many processes, RA has a particular influence over the maturation of mesoderm and somitic tissue (*Hamade et al., 2006*; *Janesick et al., 2018*; *Janesick et al., 2014*; *Li et al., 2015*). While RA is notably involved in ensuring the bilateral symmetry of the somites, it is ultimately dispensable for axis elongation in the zebrafish (*Berenguer et al., 2018*; *Bernheim and Meilhac, 2020*; *Hamade et al., 2006*; *Kumar and Duester, 2014*). Based on the timing of somite-derived RA signaling activity relative to when angioblasts migrate to the midline, we speculated it may be involved in midline angioblast migration.

Here, we show that RA-mediated somite maturation is required for the proper formation of the axial vasculature through a nonautonomous role in inducing morphological changes in surrounding tissues. A process we call notochord-endoderm separation (NES) occurs prior to angioblast migration, wherein the dorsal translocation of the notochord during development leads to its separation from the endoderm along the dorsal-ventral axis to generate a transient cavity we refer to as the ventral midline cavity (VMC). The angioblasts migrate toward and eventually into the VMC after NES. The induction of NES and the VMC is somite maturation dependent, and a delay or failure in NES can cause systemic angioblast migration defects. This evidence places somite maturation as a critical event that is required for NES and the development of the axial vasculature.

## Results

### Retinoic acid promotes convergence of angioblasts to the midline and the formation of the axial vasculature

To determine if RA signaling impacts angioblast migration, we made time-lapse videos of *tg(kdrl:GFP)* embryos, which express GFP in angioblasts (*Jin et al., 2005*). Embryos were mounted at the 10-somite stage and their migration was observed over a 3 hr time period. Wild-type angioblasts migrated to the midline in a manner consistent with previously described research (*Figure 1A*, *Video 1*; *Helker et al., 2015*; *Jin et al., 2005*; *Kohli et al., 2013*). To determine what effect the loss of RA would have on the migrating angioblasts, we generated *aldh1a2* mutants with the *tg(kdrl:GFP)* angioblast reporter in the background. Time-lapse imaging of these embryos shows defects in angioblast migration to the midline (*Figure 1B*, *Video 2*), which was also confirmed by in situ hybridization (*Figure 1— figure supplement 1*). Three other methods of disrupting RA signaling caused the same phenotype. Loss of RA by treatment with N,N-diethylaminobenzaldehyde (DEAB) (an inhibitor of the Aldh family, including Aldh1a2; *Morgan et al., 2015*), the small-molecule retinoic acid receptor (RAR) alpha and gamma inhibitor (BMS453), or expression of a dominant negative RARa using the *Tg(hsp70l:eGFP-dnHsa.RARA)* transgenic line (hereafter referred to as *HS:dnRAR*) all caused midline migration defects (*Brilli Skvarca et al., 2019*; *Figure 1—figure supplement 1A–E*). Angioblasts in these embryos show delayed migration with disorganized anterior to posterior processivity. To determine if addition of RA accelerated angioblast migration as well as anterior to posterior processivity, we compared the angioblast migration patterns of 10-somite stage *tg(kdrl:GFP)* embryos treated with either DMSO (vehicle) or 0.1 µM RA at the tailbud stage (*Figure 1C and D*, respectively). Angioblasts in the DMSO-treated embryos migrate normally, but the angioblasts treated with RA had prematurely initiated migration and arrived earlier to the midline during development (*Figure 1C and D*, respectively, *Video 3*). To determine the rate at which angioblasts reach the midline, we quantified the percentage of GFP fluorescence at the midline in relation to the total amount of fluorescence for the both wild-type and *aldh1a2-/-*. We defined angioblast fluorescence as reaching the midline if they are settled beneath the notochord. The *aldh1a2-/-* embryos show very low percentage fluorescence at the midline, relative to total fluorescence, compared to wild-type embryos 4 hr after observation, when angioblast migration is largely complete (*Figure 1E*, *Figure 1—source data 1*). On the other hand, fluorescence was higher at the midline in embryos treated with RA than in DMSO-treated control embryos (*Figure 1F*, *Figure 1—source data 2*). This shows that RA is necessary and sufficient for promoting angioblast migration to the midline. To determine at which stage RA is needed for angioblasts to migrate to the midline, we administered DEAB at different stages of development. Inhibition of RA at tailbud stage caused the angioblast migration defect, but treatment at the five-somite stage resulted in normal migration (*Figure 1H and I*, respectively), thus establishing the critical developmental window for RA signaling regulation of midline angioblast migration.

### Retinoic acid is required cell-nonautonomously for angioblast migration

To determine if RA functions cell-autonomously to induce angioblast migration, we utilized the transgenic zebrafish line *Tg(hsp70l:id3-2A-NLS-KikGR),* which carries a heat shock-inducible construct that overexpresses *id3* after a temperature shift (*Row et al., 2018*). We previously showed that overexpression of *id3* from this line can cause transplanted cells from a donor embryo to faithfully adopt an endothelial fate in a wild-type host embryo (*Row et al., 2018*). We utilized this by crossing *Tg(hsp70l:id3-2A-NLS-KikGR)*, hereafter referred to as *HS:id3*, to *HS:dnRAR*, to generate donor embryos wherein transplanted cells would be targeted to the endothelium and express dominant negative retinoic acid receptors (RARA). Cells positive for both transgenic constructs were transplanted into *tg(kdrl:eGFP)* host embryos, and these cells exhibited normal migration to the midline (*Figure 2A*, *Video 4*). In addition, we utilized an RA reporter line, *tg(RDBD,5XUAS:GFP),* to determine which tissues were subject to RA signaling (*Mandal et al., 2013*). We performed immunohistochemistry against GFP and the transcription factor Etv2, which labels the angioblasts (*Figure 2B*; *Sumanas et al., 2005*). The GFP expression indicates tissues that have been exposed to RA. These tissues included the adaxial region of the somites, the notochord, and epidermis (*Figure 2B*, white arrow, white arrowhead, and gold arrow, respectively). The primitive angioblasts, labeled with anti-Etv2 staining, do not overlap with GFP staining (*Figure 2B*, red arrows). Together, these results indicate that angioblasts are not

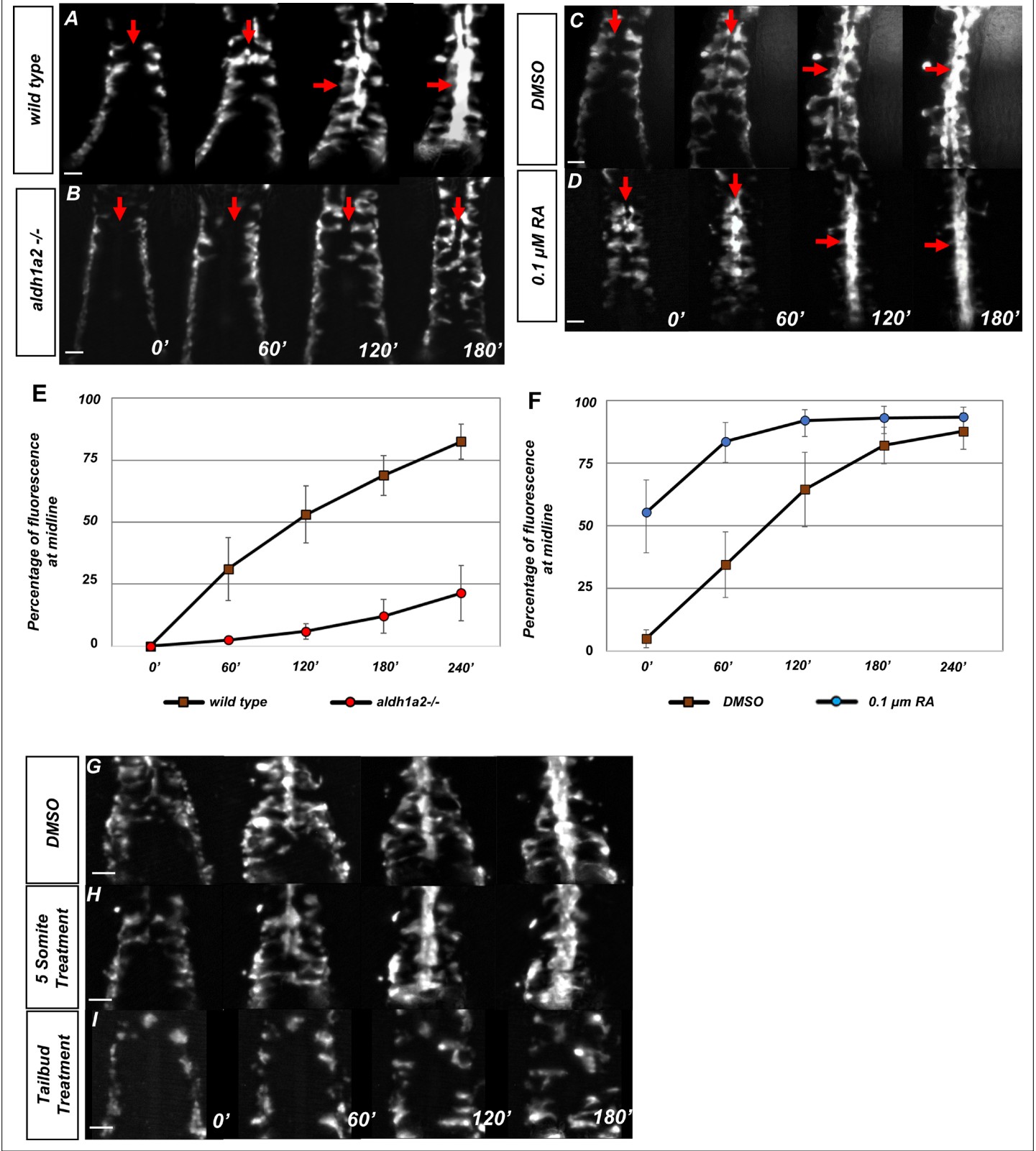

**Figure 1.** Retinoic acid (RA) is required prior to segmentation for angioblast migration. (**A–D**) Dorsal view of *tg(kdrl:eGFP)* embryos with representative images taken every 60 min over the course of 180 min, in (**A**) untreated (N = 12, see *Video 1*), (**B**) *aldh1a2-/-* mutants (N = 12, see *Video 2*), (**C**) DMSO-treated (N = 10), or (**D**) 0.1 µM RA-treated embryos (N = 8, see *Video 3*). The anterior of the embryo is at the top, and red arrows indicate the midline. (**E, F**) Graph of fluorescence intensity as angioblasts arrive at the midline for (**E**) wild-type and *aldh1a2-/-* and for (**F**) DMSO and 0.1 µM RA. Y-axis

*Figure 1 continued on next page*

*Figure 1 continued*

indicates the percentage of fluorescent intensity at midline relative to the fluorescence intensity of whole embryo. Time is measured on the X-axis. N = 8 for all conditions, error bars indicate standard deviation. (**G–I**) Time-lapse images of *tg(kdrl:eGFP)* embryos treated with either (**G**) DMSO vehicle (n = 10) or (**H**) 20 µM DEAB at the five-somite stage (n = 6), and (**I**) 20 µM DEAB at tailbud stage (n = 14). Scale bars, 50 µm.

The online version of this article includes the following source data and figure supplement(s) for figure 1:

**Source data 1.** Wild-type and *aldh1a2* -/- midline fluorescence percentages.

**Source data 2.** DMSO and RA treatment midline fluorescence percentages.

**Figure supplement 1.** Three alternative methods of retinoic acid (RA) signaling disruption inhibit midline angioblast migration.

receiving an RA signal and do not require RA signaling for migration when surrounded by a wild-type environment.

We investigated which tissue was responsible for the angioblast migration defect based on the observed activity of the reporter. Given its proximity to the midline, we chose to test whether the notochord or somitic mesoderm contributed to the defect. To test notochord contribution, we transplanted cells from a wild-type donor to a *tg(kdrl:eGFP)* host embryo to target the midline progenitors (*Row et al., 2016*). Host embryos with wild-type cells transplanted to the notochord displayed normal angioblast migration to the midline (*Figure 2E*). Similarly, transplants of *tg(HS:dnRAR)* cells into the notochord showed normal midline angioblast migration (*Figure 2F*). In addition, in situ hybridizations against a known notochord-secreted angioblast chemoattractant *apela,* as well as the receptors *aplnra* and *aplnrb,* show little change in RA-depleted conditions (*Figure 2—figure supplement 1A–F*). This indicates that the lack of RA signaling in notochord cells is not causing the angioblast migration defects. We then transplanted cells to target the somitic mesoderm. Wild-type cells transplanted into the somitic mesoderm showed normal angioblast migration to the midline (*Figure 2G*). However, when we transplanted *tg(HS:dnRAR)* cells into the somitic mesoderm, angioblasts showed migratory defects in the region of the host embryo in which they were transplanted (*Figure 2H*). Quantification of fluorescence percentage at the midline, in the chimeric regions of the notochord, showed no statistically significant difference compared to controls (*Figure 2I*, *Figure 2—source data 1*). However, quantification of fluorescence percentage in the chimeric regions of the somites showed statistical significance between wild-type and *tg(HS:dnRAR)* (*Figure 2J*, *Figure 2—source data 2*). This indicates that the somites are the principal tissue required for RA signaling-mediated midline angioblast migration.

## The somitic mesoderm is required for angioblast migration to the midline

To follow up the somite and notochord-targeted transplant results, we examined midline angioblast migration in embryos where these tissues are absent. Previous studies have investigated genes linking the notochord to angioblast migration. Mutants of the genes *noto* and *apela* were found to

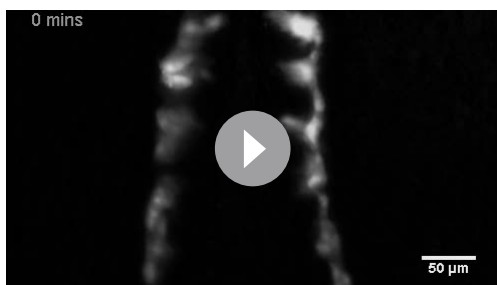

**Video 1.** Time-lapse fluorescent imaging of *tg(kdrl:eGFP)* embryo at the 10-somite stage.*tg(kdrl:eGFP)* marks angioblasts as they migrate to the midline. Angioblasts display the anterior posterior processivity while coalescing at the midline. Frame rate = 1 image/5 min. Run time = 235 min.
https://elifesciences.org/articles/74821/figures#video1

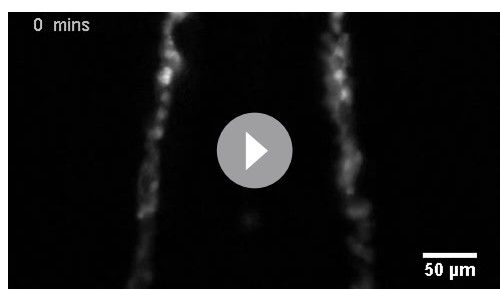

**Video 2.** Time-lapse fluorescent imaging of *tg(kdrl:eGFP), aldh1a2-/-* embryo at the 10-somite stage.*tg(kdrl:eGFP)* marks angioblasts as they migrate to the midline. Angioblasts lose anterior to posterior processivity and show disrupted migration. Frame rate = 1 image/5 min. Run time = 235 min.
https://elifesciences.org/articles/74821/figures#video2

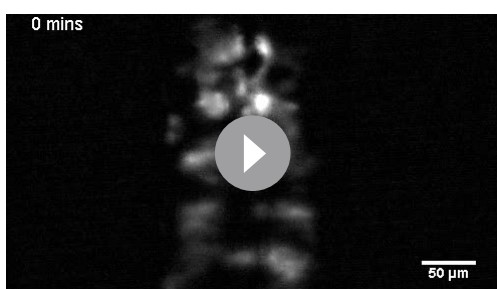

**Video 3.** Time-lapse fluorescent imaging of a 10-somite stage *tg(kdrl:eGFP)* embryo with addition of 0.1 μM retinoic acid (RA) at the tailbud stage. *tg(kdrl:eGFP)* marks angioblasts as they migrate to the midline. Angioblasts accelerate their processivity toward the midline. Frame rate = 1 image/5 min. Run time = 235 min.

https://elifesciences.org/articles/74821/figures#video3

be notochord-specific genes required for proper angioblast migration (*Cleaver and Krieg, 1998*; *Fouquet et al., 1997*; *Helker et al., 2015*). The *noto* gene induces notochord, and the notochord in turn secretes Apela, which attracts the angioblasts to the midline. Noto is required for the formation of the notochord, and in the absence of Noto, the midline cells adopt a somitic fate (*Talbot et al., 1995*).

However, it is not clear if midline convergence was delayed or absent in notochord-less embryos. In *noto* mutants, some angioblasts reach the midline at later developmental stages and resolve into a single blood vessel (*Fouquet et al., 1997*). We injected a *noto* morpholino that faithfully phenocopies the *noto* mutant into *tg(kdrl:GFP)* embryos (*Ouyang et al., 2009*). Control morphant embryos show normal angioblast migration (*Figure 3A*), whereas *noto* morphants show slowed angioblast migration (*Figure 3B*). However, midline convergence still occurs in both conditions (*Figure 3C*). This implies that the notochord is ultimately dispensable for the formation of the VMC and midline positioning of angioblasts.

Having ruled out the role of notochord tissue, we next investigated whether the somitic mesoderm is required for NES and the formation of the VMC. Previous studies have shown that the t-box transcription factor *tbx16* is required for proper somite formation (*Amacher et al., 2002*; *Goto et al., 2017*; *Griffin et al., 1998*; *Kimmel et al., 1989*; *Manning and Kimelman, 2015*; *Row et al., 2011*). In the absence of Tbx16, cells that would normally join the paraxial mesoderm fail to do so, causing a deficiency in the trunk somitic mesoderm, but leaving the endothelium intact (*Thompson et al., 1998*). We utilized *tbx16* mutants to determine whether the lack of somites caused angioblast migration defects, similar to RA loss of function. We examined midline angioblast migration in *tbx16* mutant embryos with the *tg(kdrl:GFP)* transgene in the background. In wild-type sibling embryos, angioblast migration progresses normally to the midline (*Figure 3D*). However, in *tbx16-/-* embryos, angioblast migration does not occur, and they remain in their positions in the lateral plate (*Figure 3E*). Fluorescence quantification indicated that angioblasts never reach the midline in any embryo (*Figure 3F*). To determine the effect of either notochord or somite loss on the definitive vasculature, we performed transverse sections of *tg(kdrl:GFP)* embryos injected with control, *noto,* and *tbx16* morpholinos and stained with DAPI. Control morpholino showed normal formation of the dorsal aorta and common cardinal vein (*Figure 3G*). Embryos injected with *noto* morpholino have angioblasts located at the midline that appear to have resolved into one blood vessel structure (*Figure 3H*). Embryos injected with *tbx16* morpholinos, however, show two distinct vessels located on either side of the notochord (*Figure 3I*). This is confirmed by in situ hybridization of the respective arterial and venous markers *cldn5b* and *dab2* in wild-type and *tbx16-/-* embryos (*Figure 3—figure supplement 1A–D*). This indicates that the somites are critical for angioblast convergence at the midline, while the notochord is ultimately dispensable.

## Retinoic acid induces a dorsal translocation of the notochord away from the underlying endoderm

Given the nonautonomous midline angioblast migration defects in RA loss-of-function embryos, we examined if RA manipulation altered the normal development of non-angioblast midline tissues. We utilized embryos labeled with a cell surface marker, *tg(ubb:lck-mNG)* (*Adikes et al., 2020*), and found the relative location of notochord at the midline changes based on RA activity. To measure this, embryos were treated with either DMSO, DEAB, or RA at the tailbud stage. The embryos were then fixed, deyolked, and imaged in the trunk at roughly the fifth somite (*Figure 4A–C*). The relative location of the notochord in DMSO-treated embryos was in close proximity to the ventral underlying endoderm at the 12-somite stage, but that distance increased by the 15-somite stage (*Figure 4A*). In

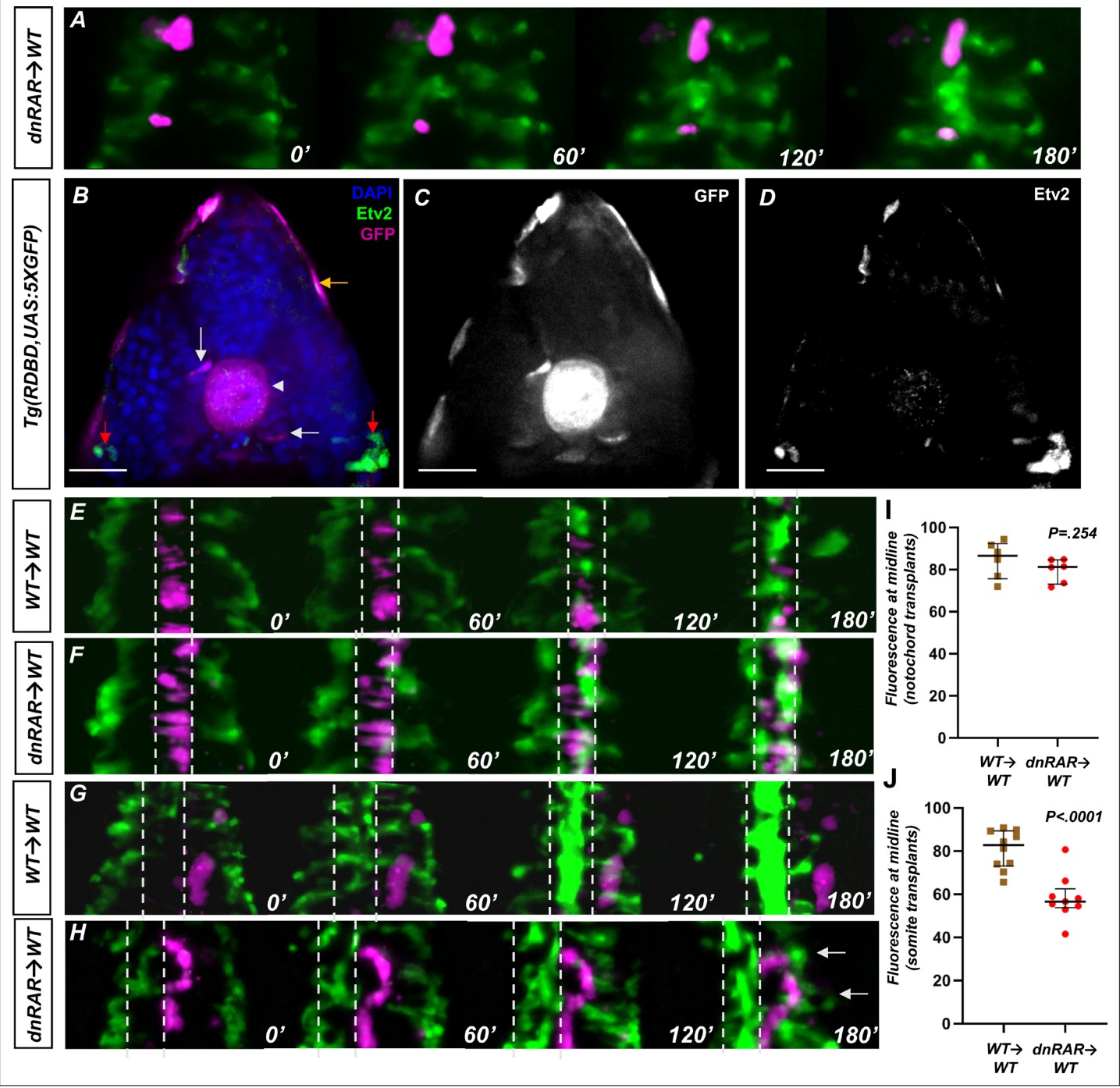

**Figure 2.** Retinoic acid (RA) signaling is not present in the endothelium and is required in the somites for angioblast migration. (**A**) Representative images of a time-lapse of *tg(HS:dnRAR), tg(HS:id3)* cells transplanted into a wild-type embryo. Cells labeled in magenta indicate migrating cells. These cells migrate along with the host angioblasts labeled by *tg(kdrl:gfp),* indicating normal migration patterns in the absence of cell-autonomous RA-depletion conditions (n = 3) (see **Video 4**). (**B**) DAPI-stained section of the 13-somite stage *tg(RDBD,5XUAS:GFP)* embryo, immunostained for GFP and Etv2 (n = 12). GFP signal is present in the adaxial region of the somite, the notochord, and the epidermis (white arrow, white arrowhead, and gold arrows, respectively). The red arrows indicate areas of Etv2 staining. (**C**) GFP channel of image in (**B**). (**D**) Etv2 channel of image in (**B**). Note the lack of overlap between Etv2 staining and GFP staining. (**E**) Representative images of a time-lapse of wild-type cells transplanted into a wild-type notochord. (**F**) Representative images of a time-lapse of *tg(HS:dnRAR)* cells transplanted into a wild-type notochord. (**G**) Representative images of a time-lapse of wild-type cells transplanted into a wild-type somite. (**H**) Representative images of a time-lapse of *tg(HS:dnRAR)* cells transplanted into a wild-type somite. White arrows indicate angioblast migration defects. (**I**) Quantification of midline fluorescence, as a percentage of total, for notochord transplants at 180'. The difference for wild-type and *tg(HS:dnRAR)* was not statistically significant (p=0.254). (**J**) Quantification of midline fluorescence,

*Figure 2 continued on next page*

*Figure 2 continued*

as a percentage of total, for somite transplants at 180′. The difference for wild-type and *tg(HS:dnRAR)* was statistically significant (p<0.0001). Scale bars, 50 µm.

The online version of this article includes the following source data and figure supplement(s) for figure 2:

**Source data 1.** Fluorescence percentage at midline for somite targeted transplants.

**Source data 2.** Fluorescence percentage at midline for notochord targeted transplants.

**Figure supplement 1.** Apela/Aplnr axis is not altered by the changes in retinoic acid signaling.

DEAB-treated embryos, the notochord remains ventrally localized at both the 12-somite stage and 15-somite stage (*Figure 4B*). When the same experiment was done with embryos exposed to exogenous RA, the notochord was prematurely dorsally localized at the 12-somite stage and extended even further by the 15-somite stage (*Figure 4C*).

We then determined the dynamics of the notochord displacement in real time. Using DIC images, we were able to overlay the lateral trunk with the notochord. We then observed the notochord displacement in a 250′ time-lapse for DMSO-treated wild-type, *aldh1a2-/-* with DMSO, and exogenous RA (*Figure 4D–F*, respectively). We quantified this data by measuring the distance of the ventral aspect of the notochord to the yolk cells of the embryo (yellow brackets). We found that *aldh1a2-/-* embryos showed delayed dorsal displacement relative to DMSO in both the timing of dorsal movement and the magnitude of that movement (*Figure 4G*, *Figure 4—source data 1*).

Previous work showed that as somites mature they extend in both the dorsal and ventral axis (*Tlili et al., 2019*), and based on this we hypothesized that this extension was concurrent with the dorsal translocation of the notochord. However, we wanted to rule out the effect of forces on the anterior-most and posterior-most regions of the notochord. The notochord is a rigid structure that is anchored in the tailbud near the notochord primordia and extends into the head mesoderm medial to the otic vesicle (*Kimmel et al., 1995*). To eliminate the contribution other tissues could have on NES, we generated trunk explants from embryos containing both *tg(actc1b:gfp)* and *tg(tbxta:kaede)* transgenes that contained roughly 10 somites of the embryo. The 10-somite stage embryos were sectioned near the somite borders of the 1st and 10th somite. We performed live imaging of these trunk explants, focusing on the immature somites of the posterior region (white boxes) (*Figure 4H–J*, *Video 5*). In order to better visualize the notochord, we separated the *tg(actc1b:gfp)* and *tg(tbxta:kaede)* signals into separate channels. This distinguished the notochord reporter *tg(tbxta:kaede)* (*Figure 4H′–J′*) from the somite reporter *tg(actc1b:gfp)* (*Figure 4H″–J″*). The notochord reporter showed clear dorsal movement away from the autofluorescent yolk (*Figure 4H′–J′*). The somites also showed the corresponding expansion along the dorsal-ventral axis (*Figure 4H″–J″*). Together, this data indicates that the proper development of the midline is both RA-dependent and requires only factors localized to the trunk of the embryo.

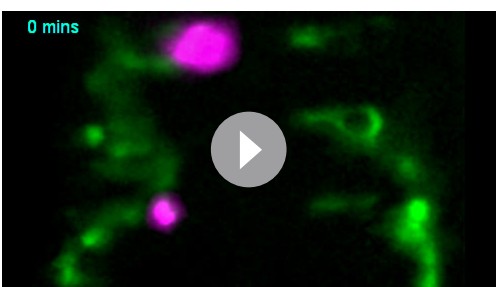

**Video 4.** Time-lapse fluorescent imaging of *HS:id3, HS;dnRAR* cells in a *tg(kdrl:eGFP)* host.
Cells expressing *HS:id3, HS;dnRAR* migrate to the midline along with angioblasts in a *tg(kdrl:eGFP)* host. Cells with dnRAR migrate faithfully with angioblasts. Frame rate = 1 image/5 min. Run time = 180 min.
https://elifesciences.org/articles/74821/figures#video4

## Notochord-endoderm separation facilitates the midline convergence of angioblasts

We speculated that RA-dependent changes at the midline could be the cause of the angioblast migration defect. In order to test this, we generated a heat shock-inducible transgenic reporter of F-actin, *tg(hsp70l:lifeact-mScarlet)*, which labels filamentous actin throughout the embryo and allows visualization of all cells (*Riedl et al., 2008*). We hereafter refer to *tg(hsp70l:lifeact-mScarlet)* as *tg(HS:lifeact)*. We then crossed this reporter to the *tg(kdrl:eGFP)* line to label the angioblasts. We treated embryos with either DMSO (*Figure 5A–C*), exogenous RA (*Figure 5D and E*), or DEAB (*Figure 5F–I*). Embryos were sectioned at the region of the fifth somite (*Figure 5A–I*).

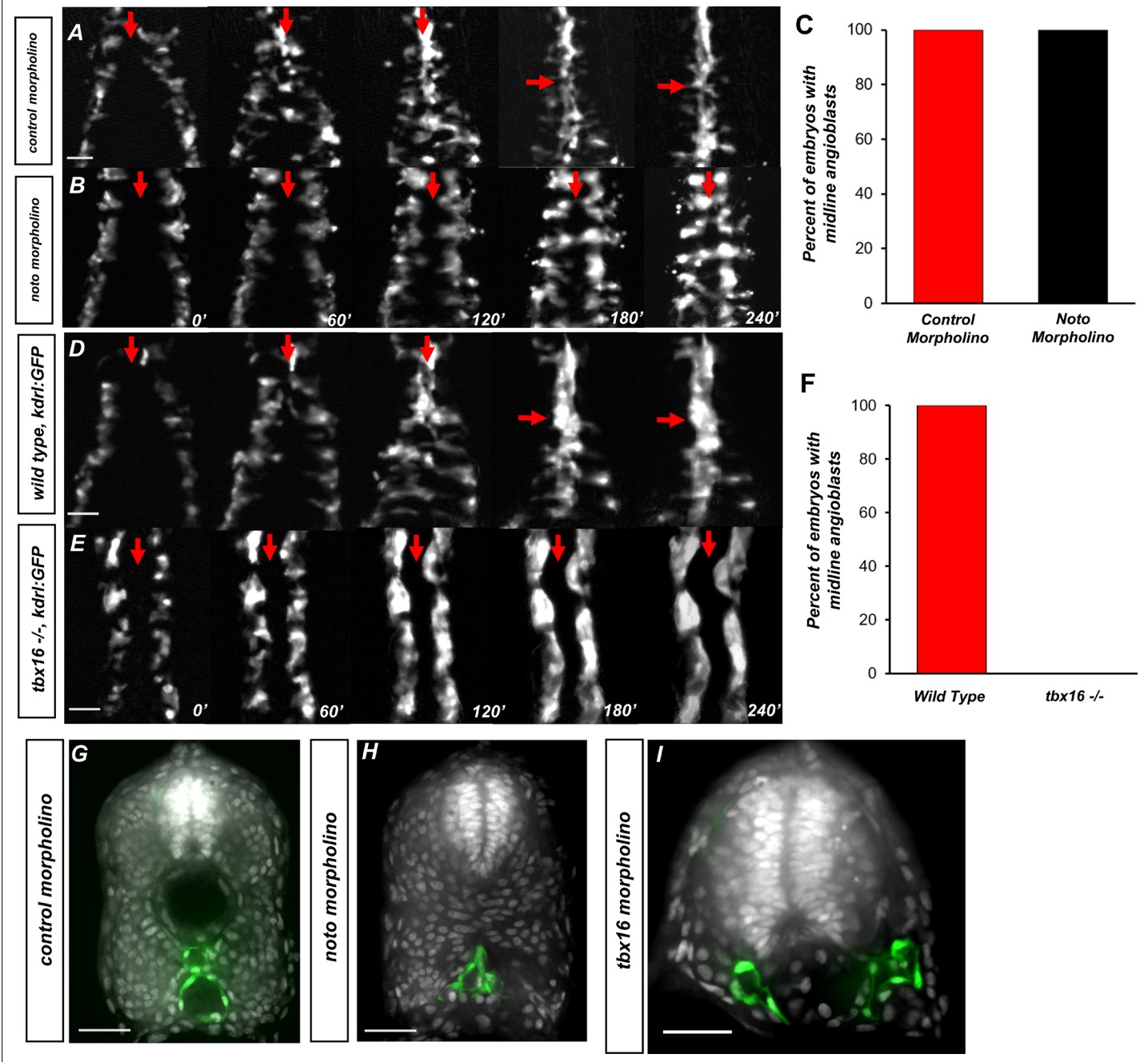

**Figure 3.** The somitic mesoderm, not the notochord, is required for midline convergence of angioblasts. (**A**) Time-lapse imaging of *tg(kdrl:eGFP)* embryos injected with control morpholino over a 240' period. (**B**) *tg(kdrl:eGFP)* embryos injected with *noto* morpholino over a 240' period. Red arrows indicate the midline. (**C**) Graph of embryos with midline angioblasts in *noto* and control morphants after 240'. N = 15 for both conditions (**D**) Time-lapse imaging of wild-type, *tg(kdrl:eGFP)* embryos over a 240' period. (**E**) Time-lapse imaging of *tbx16-/-*, *tg(kdrl:eGFP)* embryos over a 240' period. Red arrows indicate the midline. (**F**) Graph of embryos with midline angioblasts in wild-type and *tbx16-/-* after 240'. N = 18 for both conditions. (**G**) Section of wild-type, *tg(kdrl:eGFP)* embryos over a 240' period. (**H**) Section of 24 hpf *tg(kdrl:eGFP)* embryos injected *noto* morpholino and stained with DAPI. (**I**) Section of 24 hpf *tg(kdrl:eGFP)* embryos injected *tbx16* morpholino. Blood vessels are labeled in green and nuclei labeled in gray. Scale bars, 50 μm.

The online version of this article includes the following figure supplement(s) for figure 3:

**Figure supplement 1.** Bifurcated angioblasts in *tbx16* mutants preferentially join the venous population.

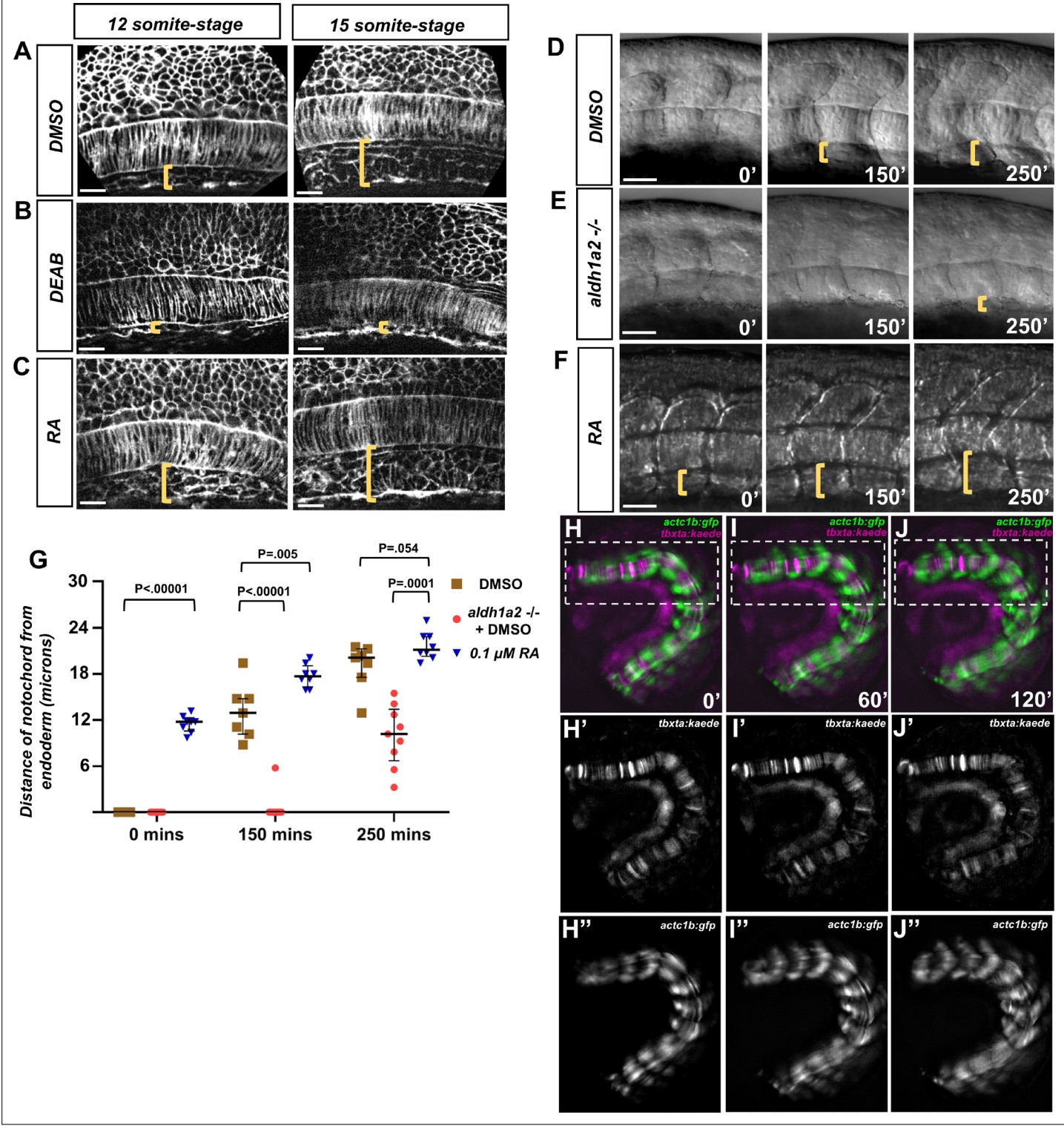

**Figure 4.** Retinoic acid (RA) signaling promotes dorsal translocation of the notochord. (**A–C**) Fluorescent images of fixed (**A**) DMSO-treated, (**B**), DEAD-treated, and (**C**) RA-treated *tg(ubb:lck-m*NG) embryo trunks at the 12- and 15-somite stage. Yellow brackets indicate the distance between the notochord and underlying yolk cell. (**D–F**) DIC time-lapse images of the notochord and zebrafish trunk. Focal planes of the notochord and somites were overlaid to show the relative location to one another. Yellow brackets indicate the distance of the ventral notochord to the ventral-most portion of the embryo in (**D**) DMSO-treated embryos, (**E**) *aldh1a2-/-* embryos treated with DMSO, or (**F**) 0.1 μM RA-treated embryos. (**G**) Quantification of dorsal translocation of notochord from the ventral tissue. Black lines indicate the median and interquartile range. Wild-type are indicated by brown squares, while *aldh1a2-/-* and 0.1 μM RA embryos are shown by red circles and blue triangles, respectively. At t = 0, DMSO vs. RA is p<0.00001. At 150',

*Figure 4 continued on next page*

*Figure 4 continued*

DMSO is p<0.00001 and p=0.005 vs. *aldh1a2-/-* and RA treatment, respectively. At 250', DMSO is p=0.0001 and 0.054 vs. *aldh1a2-/-* and RA treatment, respectively. (**H–J**) Time-lapse image of a trunk explant from *tg(actc1b:gfp)*, *tg(tbxta:kaede)* embryo at (**H**) 0', (**I**) 60', and (**J**) 120'. (**H'–J'**) Time-lapse image of explant showing *tg(tbxta:kaede)* only. (**H"–J"**) Time-lapse image of explant showing *tg(actc1b:gfp)* only. Time-lapse shows the notochord adjacent the yolk at time 0', but moves dorsally from the yolk at time 120'. Scale bars, 25 µm.

The online version of this article includes the following source data for figure 4:

**Source data 1.** Distances of notochord to endoderm after RA manipulation.

Embryos that were treated with DMSO showed normal midline angioblast migration patterns from the 12-somite stage, 15-somite stage, and 18-somite stage (*Figure 5A–C*). As angioblasts approach the midline, a gap appears between notochord/hypochord and the underlying endoderm (*Figure 5A*). This gap is acellular as a 15 µm max projection shows an absence of both nuclei and F-actin labeling (*Figure 5—figure supplement 1A–C*). This gap is filled by angioblasts by the 15-somite stage, separating the notochord from the endoderm (*Figure 5B*). We call the process of dorsal translocation of the notochord the notochord-endoderm separation (NES). The transient opening that forms from NES is referred to as the ventral midline cavity (VMC). As angioblast migration continues, additional angioblasts occupy the VMC until the 18-somite stage when angioblast migration is largely complete for the fifth somite region of the trunk (*Figure 5C*).

To determine if activation of RA is sufficient to accelerate angioblast migration and NES concurrently, we treated embryos with exogenous RA and sectioned them at the 12-somite and 15-somite stage. Embryos sectioned at the 12-somite stage showed premature angioblast migration to the midline and NES, indicating that angioblast migration and NES are responsive to exogenous RA signal (*Figure 5D*). Angioblasts continue to be at the midline at the 15-somite stage (*Figure 5E*). The 18-somite stage embryos were not available as posterior elongation discontinued in prolonged RA exposure, as previously shown (*Martin and Kimelman, 2010*). To determine whether angioblast migration and NES were attenuated in RA-depletion conditions, embryos were treated with DEAB and sectioned at the 12-somite, 15-somite, 18-somite, and 21-somite stages (*Figure 5F–I*, respectively). At the 12-somite stage, the angioblasts are localized in the lateral plate mesoderm (*Figure 5F*). Focusing on the midline of the embryo, we observe that the notochord remains flush against the endoderm with no visible VMC or NES (*Figure 5G*). At the 15-somite stage, we observe some midline movement of the angioblasts; however, the notochord remains flush against the endoderm and no angioblasts have been able to reach the midline (*Figure 5G*). During the 18-somite stage, a small VMC has formed between the notochord and endoderm. The leading angioblasts have partially migrated into this VMC; however, the bulk of the angioblasts still reside outside of the midline (*Figure 5H*). By the 21-somite stage, the angioblasts have largely resolved to the midline, although with some angioblasts still residing outside the midline (*Figure 5I*).

Given the pronounced delays in NES formation for anterior regions in RA-depleted embryos, we suspected that NES was a consequence of the known processive somite maturation that begins in the anterior regions and progresses posteriorly (*Stern and Piatkowska, 2015*). We sectioned 15-somite stage *tg(HS:lifeact)*, *tg(kdrl:eGFP)* embryos in the anterior and posterior regions corresponding to the 5th and 12th somite (*Figure 5J*). In more immature 12th somite, angioblasts reside in their lateral regions and NES is not pronounced (white dashed lines outline the somites, and yellow brackets show the distance from the notochord to the ventral endoderm) (*Figure 5K*). In the fifth somite of the same embryo, angioblasts reside at the midline and NES is very pronounced, indicating that NES occurs anterior to posteriorly (*Figure 5L*). The lack of angioblasts at the 12th somite, and low NES, is similar to the DEAB-treated embryos sectioned at the more mature 5th somite position (*Figure 5M*).

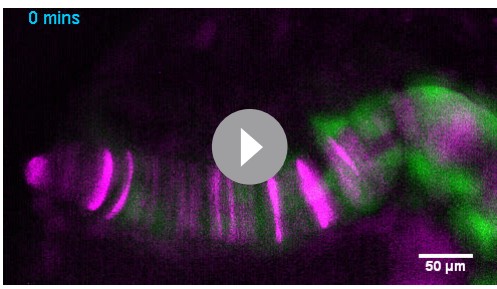

**Video 5.** Time-lapse fluorescent imaging of *tg(tbxta:kaedae)*, *tg(actc1b:gfp)* trunk explant. Trunk explant shows notochord displacement away from ventral-most portion of somites. Frame rate = 1 image/15 min. Run time = 285 min.

https://elifesciences.org/articles/74821/figures#video5

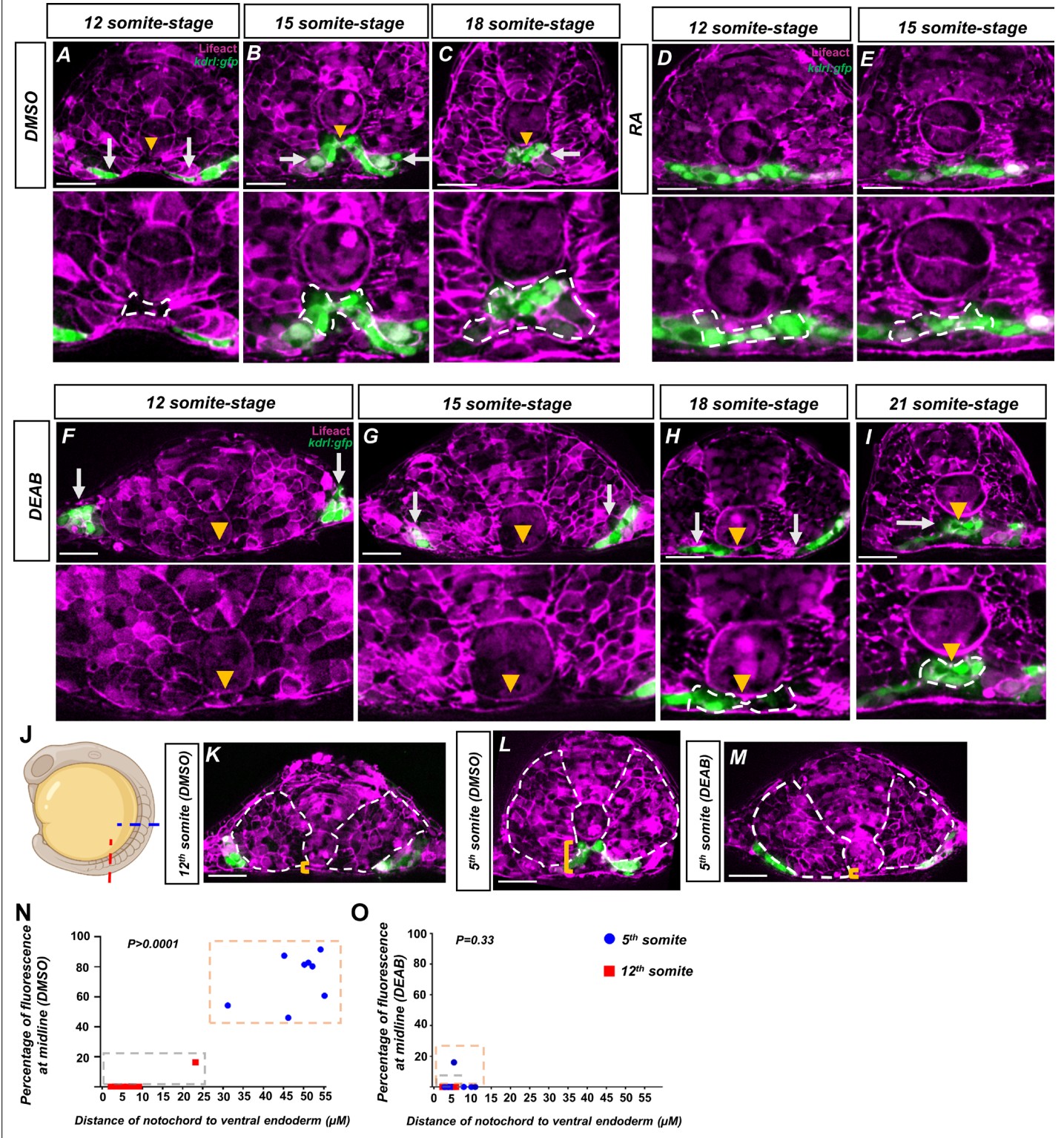

**Figure 5.** Retinoic acid (RA) mediates a notochord-endoderm separation required for terminal angioblast migration. (**A–I**) Embryos generated by crossing *tg(HS:lifeact)* to *tg(kdrl:eGFP)* to label both actin and angioblasts. Yellow arrowheads indicate the midline while white arrows indicate angioblasts. A magnified image includes a white dashed line to indicate notochord-endoderm separation. (**A–C**) DMSO-treated embryos sectioned at the fifth somite during the (**A**) 12-somite stage (n = 8), (**B**) 15-somite stage (N = 8), and (**C**) 18-somite stage (N = 6). (**D, E**) 0.1 μM RA-treated embryos sectioned at the fifth somite during the (**D**) 12-somite stage (N = 8) and (**E**) 15-somite stage (n = 8). (**F–I**) 20 μM DEAB-treated embryos sectioned at the fifth somite during the (**F**) 12-somite stage (N = 7), (**G**) 15-somite stage (N = 8), (**H**) 18-somite stage (N = 6), and (**I**) 21-somite stage (N = 6). (**J–**

*Figure 5 continued on next page*

Figure 5 continued

L) 15-somite stage embryos generated by crossing *tg(HS:lifeact)* to *tg(Kdrl:eGFP)*. White dashed lines indicate outline of somites while yellow brackets indicate distance of notochord to ventral endoderm. (**J**) Schematic showing the sectioned region for experiments in (**K–O**). Red line indicates sectioning at 5th somite, and the blue line indicates sectioning at the 12th somite. (**K**) A DMSO embryo sectioned at the 12th somite. Note the lack of midline angioblasts and small distance between the notochord and the ventral endoderm. (**L**) A DMSO-treated embryo sectioned at the fifth somite. Note the angioblasts at the midline and larger distance from the notochord to the ventral endoderm. (**M**) A DEAB embryo sectioned at the fifth somite. Note the lack of midline angioblasts and small distance between the notochord and the ventral endoderm. (**N**) A two-variable graph showing the percentage of angioblast fluorescence at the midline compared to the distance between the notochord and the ventral endoderm. Blue dots indicate DMSO-treated, 15-somite stage embryos sectioned at the 5th somite, and red squares are the same embryos sectioned at the 12th somite (N = 8). (**O**) A two-variable graph showing the percentage of angioblast fluorescence at the midline compared to the distance between the notochord and the ventral endoderm. Blue dots indicate DEAB-treated, 15-somite stage embryos sectioned at the 5th somite, and red squares are the same embryos sectioned at the 12th somite (N = 8). Scale bars, 50 μm.

The online version of this article includes the following source data and figure supplement(s) for figure 5:

**Source data 1.** Fluorescence percentage at midline for DMSO treated embryos at 5th and 12th somite.

**Source data 2.** Fluorescence percentage at midline for DEAB treated embryos at 5th and 12th somite.

**Figure supplement 1.** The ventral midline cavity is acellular.

To quantify how the dorsal translocation of the notochord was correlated with angioblasts arriving to the midline, we created a two-variable graph to measure both the distance of the notochord to the ventral endoderm and the percentage of angioblasts arriving at the midline for a given somite. For this analysis, we used 15-somite stage embryos sectioned at the 5th somite and 12th somite. We expect that immature somites in the posterior would have little midline migration of angioblasts and the anterior, with more mature somites, would have significant midline migration. We measured the distance in microns on the x-axis and the percentage of midline angioblasts on the y-axis (*Figure 5N and O*). In DMSO-treated embryos, embryos sectioned at the fifth somite grouped together, showing both significant notochord-endoderm distance and angioblasts residing at the midline (*Figure 5N*, blue dots in the orange box, *Figure 5—source data 1*). Embryos sectioned at the 12th somite show short notochord-endoderm distance and little angioblast migration towards the midline (*Figure 5N*, red squares in gray box). When comparing angioblast migration, the 5th somite and 12th somite regions were significantly different ($p > 0.0001$). The corresponding sectioning of DEAB-treated embryos shows a difference compared to wild-type in the fifth somite region. Embryos sectioned at the fifth somite have a small distance from the notochord to endoderm, as well as no angioblast migration to the midline (*Figure 5O*, blue dots in orange box, *Figure 5—source data 2*). Interestingly, the notochord-endoderm distance/migration percentage ratio strongly resembled the more immature somites. The angioblast migration percentage was not statistically significant between the 5th and 12th somite in RA-depletion conditions ($p = 0.33$). This indicates that the midline exists in a more immature state in RA-depleted conditions, and the more mature state provides space in which the angioblasts can migrate into.

## Retinoic acid induces changes in the definitive vasculature

Pharmacological inhibition of RA signaling has previously been shown to effect the zebrafish vasculature, notably causing a smaller than normal dorsal aorta (*Pillay et al., 2016*). To confirm this effect in the *aldh1a2-/-* mutant, we sectioned 24 hpf *tg(kdrl:GFP)* and *aldh1a2-/-* embryos, stained the sections with DAPI, and imaged them using spinning disk confocal microscopy (*Figure 6A and B*). Images show that the dorsal aorta and posterior cardinal vein in wild-type embryos are roughly equivalent in size (*Figure 6A*, white arrows). However, in the *aldh1a2-/-* embryos, the size of the dorsal aorta is reduced but remained lumenized (*Figure 6B*, white arrows). We confirmed this observation by in situ hybridization of the arterial marker *cldn5b* and the venous marker *dab2*, which showed similar effects (*Figure 6—figure supplement 1A–D*; *Casie Chetty et al., 2017*). To determine if this trend extended beyond the 24 hpf developmental stage, we imaged the trunk of 5 dpf *tg(kdrl:GFP)* embryos along with *tg(kdrl:GFP)*, *aldh1a2-/-* siblings (*Figure 6C and D*). The *tg(kdrl:GFP)* embryos alone showed no vascular defects and normal dorsal aorta and posterior cardinal vein (*Figure 6C*, white arrow and yellow arrowhead, respectively). However, the vasculature of the *aldh1a2-/-* siblings shows a large reduction of the size of the dorsal aorta (*Figure 6D*, white arrow). In addition, it appears that the posterior cardinal vein of *aldh1a2-/-*increased in size compared to its sibling (*Figure 6D*, yellow arrowhead).

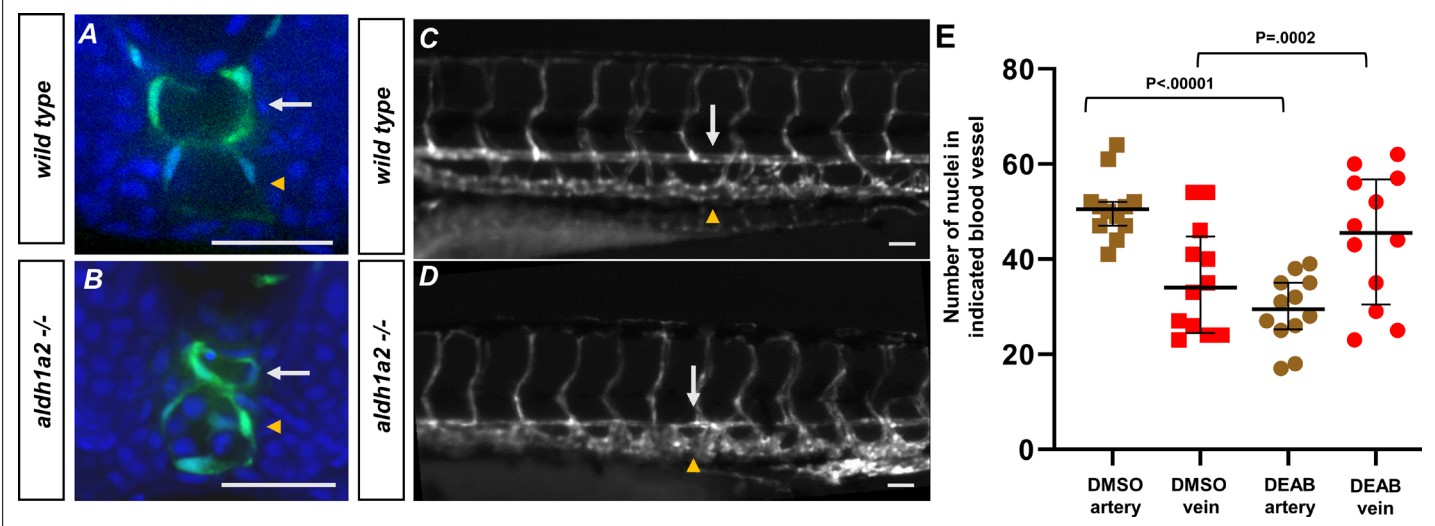

**Figure 6.** Retinoic acid loss contributes to changes in the definitive vasculature, resulting in large veins and small arteries. (**A, B**) A 24 hpf *tg(kdrl:eGFP)* embryo labeled with DAPI. (**A**) Wild-type embryos show lumenized blood vessels and normal-sized dorsal aorta. (**B**) Labeled *aldh1a2-/-, tg(kdrl:eGFP)* sibling shows small lumenized dorsal aorta with large posterior cardinal vein. (C) A 5 dpf *tg(kdrl:eGFP)* embryo. White arrows indicate the dorsal aorta, and yellow arrowheads indicate the cardinal vein. (D) A 5 dpf *tg(kdrl:eGFP), aldh1a2-/-* embryo. White arrows indicate reduced size of dorsal aorta. (**E**) Quantification of DMSO (n = 12) or 20 µM DEAB (n = 12)-treated *tg(kdrl:nls-eGFP)* embryos. Nuclei were counted in each structure over a 225 µM long transverse section of the zebrafish trunk. Two-tailed p-values for the unpaired *t*-test of wild-type vs. 20 µM DEAB-treated embryos are <0.00001 and 0.0002 for arteries and veins, respectively. Scale bars, 50 µm.

The online version of this article includes the following source data and figure supplement(s) for figure 6:

**Source data 1.** Nuclei count in dorsal aorta and cardinal vein.

**Figure supplement 1.** Effects of retinoic acid (RA) depletion on arterial and venous markers.

To determine if the size change was the result of a reduced cell number in the DA in response to RA loss, we utilized the transgenic line *tg(kdrl:nls-GFP)*, which labels the nucleus of endothelial cells to count the number of cells in each blood vessel over a 225 µM horizontal section of a 24 hpf zebrafish embryo. Embryos were treated with either DMSO or with DEAB to inhibit RA. We see reduced cell number in the DEAB-treated artery versus the DMSO-treated artery (*Figure 6E*, *Figure 6—source data 1*). This corresponds to an increase in the number of cells in the veins during DEAB treatment compared to DMSO. While we cannot definitively say that the migration defects caused the arterial reduction, it indicates that RA is required for the proper formation of the definitive vasculature.

## Retinoic acid-induced somite morphogenesis facilitates notochord-endoderm separation and facilitates midline angioblast migration

While NES and midline formation were shown to be highly correlated, we previously showed that cell-autonomous RA depletion within the somite caused angioblast migration defects. Therefore, we sought to determine if RA depletion within the somites caused morphological changes that compromised NES and angioblast migration to the midline. During normal development, newly born somites initially lay flat and extend farther in the medial-lateral axis, demonstrated by a section of 15-somite stage embryo at the 12th somite (*Figure 7A*, white dashed lines; *Tlili et al., 2019*). However, as somites mature, they will narrow in the medial-lateral axis and extend in the dorsal-ventral axis, as demonstrated by a section of the fifth somite of a 15-somite stage embryo (*Figure 7B*, white dashed line; *Tlili et al., 2019*). We therefore sought to determine if somite maturation could be affected cell-autonomously in RA-depletion conditions. We therefore transplanted either wild-type or *HS:dnRAR* cells into *tg(ubb:lck-m*NG) embryos. Embryos were then sectioned at roughly the fifth somite region for 12-somite stage embryos (*Figure 7C and D*). Embryos that were transplanted with wild-type cells showed little difference with the contralateral somite in terms of shape and maturation (*Figure 7C*, white dashed lines). However, embryos transplanted with *HS:dnRAR* cells showed a defect in the somite containing donor cells, with the somite extending farther in the medial-lateral axis while the

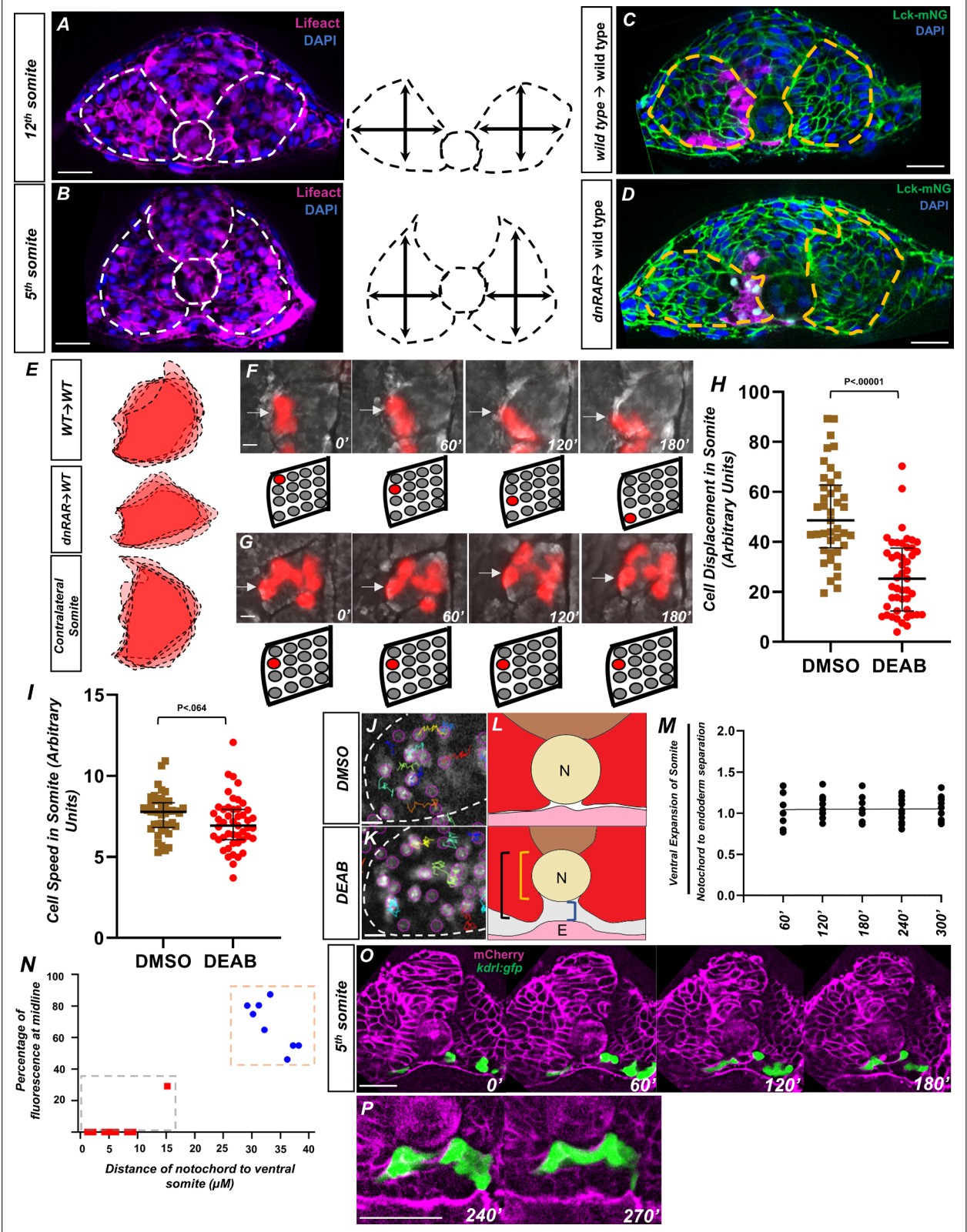

**Figure 7.** Retinoic acid controls intrasomitic cellular movements and somite shape changes that contribute to angioblast convergence to the midline. (**A**) Image and schematic of newly born 12th somite using *tg(HS:lifeact), tg(kdrl:e*GFP) embryo as reference. Dashed lines indicate the somite and notochord. (**B**) Schematic of more mature fifth somite (at the 12-somite stage) of *tg(HS:lifeact), tg(kdrl:e*GFP) embryo as reference. Dashed lines indicate somite and notochord. (**C**) Transplant of rhodamine dextran-labeled wild-type cells into *tg(ubb:lck-m*NG). Dashed lines indicate somite shape.

*Figure 7 continued on next page*

*Figure 7 continued*

(**D**) Transplant of *tg(HS:dnRAR)* cells into *tg(ubb:lck-mNG)*. Dashed lines indicate somite shape. Note the immature shape of the somite containing transplanted cells. (**E**) Overlapping somite shapes of either WT→WT, *tg(HS:dnRAR)*→WT, or the contralateral somite *tg(HS:dnRAR)*→WT. (**F, G**) Time-lapse images of transplanted rhodamine dextran-labeled cells in DMSO and DEAB embryos at 12-somite stage. (**F**) DMSO-treated embryos over 180′. White arrows indicate labeled cells. (**G**) 20 μM DEAB-treated embryos over 180′. Scale bars, 50 μm. Retinoic acid-depleted embryos show little cellular movement during somite development. (**H**) Quantification of nuclei displacement in the second somite of DMSO and DEAB embryos over a 4 hr time period. Displacement is defined as the distance between the beginning location and end location of the nuclei within the somite 2. (**I**) Quantification of speed in somite 2, defined as average units over 4 hr in 10 min increments. (**J**) Sample of tracks from DMSO-treated *tg(hsp70l:CAAX-mCherry-2A-NLS-KikGR)* embryos. (**K**) Sample tracks of DEAB-treated *tg(hsp70l:CAAX-mCherry-2A-NLS-KikGR)* embryos. Tracks sort from red to blue, with red being the highest displacement. Scale bars, 20 μm (**L**) Diagram of ventral somite expansion and notochord to endoderm separation. Ventral somite expansion is calculated by ventral somite length (black bracket) – the notochord diameter (yellow bracket). Notochord/endoderm separation is shown in the blue bracket. (**M**) Quantification of the ventral expansion of the somite over notochord/endoderm separation in wild-type embryos. The linear regression indicated by the black line, where slope = 0.002, shows ventral somite expansion correlates with notochord/endoderm separation. (**N**) A two-variable graph showing the percentage of angioblast fluorescence at the midline compared to the distance of ventral somite expansion beneath the notochord. Blue dots indicate DMSO-treated, 15-somite stage embryos sectioned at the 5th somite, and red squares are the same embryos sectioned at the 12th somite (N = 8). Scale bars, 50 μm. (**O**) Live section of *mCherry*-CAAX-injected *tg(kdrl:eGFP)* embryos over 120′. Yellow arrowheads indicate the midline while white arrows indicate angioblasts. (**P**) Same live section as in (**N**), except at 240′ focused at the midline. Note the separation of the hypochord and endoderm prior to midline fusion of angioblasts. Scale bars, 50 μm.

The online version of this article includes the following source data and figure supplement(s) for figure 7:

**Source data 1.** Displacement of nuclei within the somite.

**Source data 2.** Speed of nuclei within the somite.

**Source data 3.** The ratio of ventral expansion of the somite over NES.

**Source data 4.** The percentage of midline fluorescence in relation to ventral somite expansion.The percentage of midline fluorescence in relation to ventral somite expansion.The percentage of midline fluorescence in relation to ventral somite expansion.

**Figure supplement 1.** Loss of retinoic acid (RA) function in chimeric embryos shows regional defects in notochord-endoderm separation (NES) and angioblast migration.

**Figure supplement 2.** Retinoic acid (RA) signaling affects the expression of genes associated with somitic movements.

---

contralateral somite extends farther in the dorsal-ventral axis (*Figure 7D*, white dashed lines). To illustrate this effect across multiple experimental embryos, we show overlapping shapes of chimeric wild-type somites, the chimeric *HS:dnRAR* somites, and the contralateral somites from *HS:dnRAR* chimeras (*Figure 7E*). The overlapping images show a consistent trend of a shallow dorsal-ventral axis in the chimeric *HS:dnRAR* somites compared to the wild-type somites. Interestingly, when analyzing NES and angioblast migration in chimeric embryos, we can see regional openings and advanced migration of angioblasts in the contralateral somites compared to the somites containing *HS:dnRAR* cells (*Figure 7— figure supplement 1B and D*). Together, this indicates that RA is required cell-autonomously for somite maturation and dorsal-ventral extension.

We speculate that the failure of somites to mature was the result of reduced cell movements within the somite. To confirm this, mosaically labeled cells were generated by transplanting rhodamine dextran-labeled cells from a wild-type donor into a wild-type host. The embryos were treated with either DMSO or DEAB and time-lapse imaged over the course of 4 hr (*Figure 7F and G*, respectively). Cells in the DMSO-treated embryos showed broad movements consistent with that of somite rotation, where cells located in the anterior of the somite will translocate to the posterior of the somite (*Hollway et al., 2007*; *Figure 7F*, white arrows). However, cells in the DEAB-treated embryos show little intratissue displacement and cells largely remained static (*Figure 7G*, white arrows). To quantify this, we utilized *tg(hsp70l:CAAX-mCherry-2A-NLS-KikGR)*

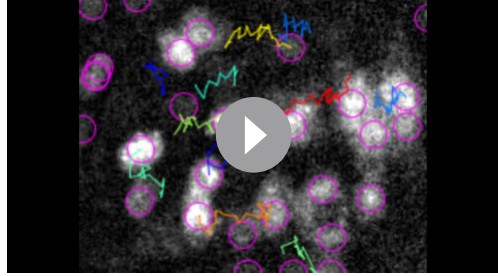

**Video 6.** Time-lapse confocal fluorescent imaging of an *HS:mCherry-CAAX-p2a-NLS-kikume* embryo in DMSO treatment. *HS:mCherry-CAAX-p2a-NLS-kikume* marks nuclei within the somite. Sample tracks ranging from red (most displacement) to blue (least displacement) show broad movement within the somite. Frame rate = 1 image/5 min. Run time = 240 min.

https://elifesciences.org/articles/74821/figures#video6

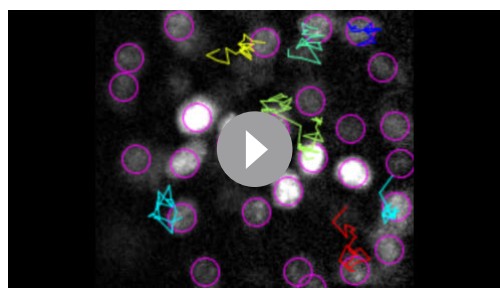

**Video 7.** Time-lapse confocal fluorescent imaging of an *HS:mCherry-CAAX-p2a-NLS-kikume* embryo in DEAB treatment. *HS:mCherry-CAAX-p2a-NLS-kikume* marks nuclei within the somite. Sample tracks ranging from red (most displacement) to blue (least displacement) show little directional movement in 20 μM DEAB treatment conditions. Frame rate = 1 image/5 min. Run time = 240 min.

https://elifesciences.org/articles/74821/figures#video7

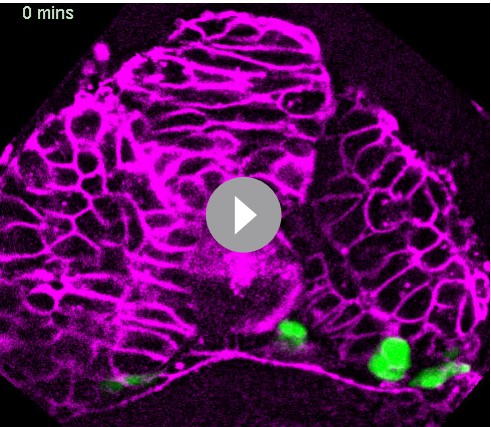

**Video 8.** Time-lapse fluorescent video of *mCherry-CAAX* mRNA-injected *tg(kdrl:eGFP)* explant at the 10-somite stage. *tg(kdrl:eGFP)* marks angioblasts as they migrate to the midline, and *mCherry-CAAX* marks the cell surfaces. A trunk explant, imaged around the fifth somite, shows normal migration of the angioblasts to the midline. Frame rate = 1 image/5 min. Run time = 280 min.

https://elifesciences.org/articles/74821/figures#video8

embryos to observe cell movement across the entirety of the somite. Embryos were heat shocked during shield stage and subsequently treated with either DMSO or DEAB and their somites were time-lapse imaged for 4 hr. Using Fiji Trackmate software (*Tinevez et al., 2017*), we quantified the displacement and speed of cell movements within the second somite (*Figure 7H and I*, *Figure 7—source data 1* and *Figure 7—source data 2*). Cells in the DMSO-treated embryos showed significant amounts of displacement throughout the somite (*Figure 7H*, *Figure 7—source data 1*). However, the DEAB cells showed less displacement over time (*Figure 7H*). Quantification of speed in the same manner showed no statistically significant difference between DMSO and DEAB treatment (*Figure 7I*, *Figure 7—source data 2*). A sample of somite cell tracks shows little directional movement in loss of RA conditions compared to DMSO, indicating that cells are not moving in a manner consistent with reshaping of the somite, as well as suggesting a role for RA signaling in somite rotation (*Figure 7J and K*, *Videos 6 and 7*, respectively). The Cxcr4a/Cxcl12a signaling axis has been shown to be responsible for controlling cell movements in the somite. In situ hybridization for *cxcr4a* shows a loss of expression in RA-depletion conditions and gain of expression in RA-addition conditions (*Hollway et al., 2007*; *Figure 7—figure supplement 2D and E*).

We speculated that the ventral expansion of the somite that occurs during somite cell rearrangement and maturation was responsible for NES. At the 10-somite stage, embryos have their notochord adjacent to the endoderm and level to ventral-most edge of the somite (*Figure 7L*). As the embryo develops, the somite expands ventrally (black bracket). As the ventral distance of the somite extends relative to the dorsal part of the notochord (yellow bracket), it induces space that the angioblasts occupy (blue brackets). Starting from the 10-somite stage, we measured the distance of ventral expansion relative to the distance between the notochord and the endoderm every hour for 4 hr. A simple linear regression of the data, indicated by the black line, shows a slope of 0.002 and y-intercept of 1.04 (*Figure 7M*, *Figure 7—source data 3*). The flat slope indicates little variance across time, and the y-intercept near 1 indicates that NES is equivalent to ventral expansion of the somite. The 1:1 correlation of ventral

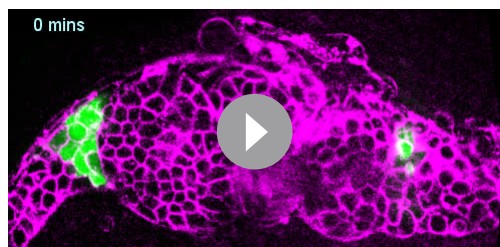

**Video 9.** Time-lapse fluorescent video of *mCherry-CAAX* mRNA-injected *tg(kdrl:eGFP)* explant at the 10-somite stage, treated with 20 μM DEAB. *tg(kdrl:eGFP)* marks angioblasts as they migrate to the midline. The trunk explant shows angioblast migration defects seen in fixed DEAB sections. Frame rate = 1 image/5 min. Run time = 175 min.

https://elifesciences.org/articles/74821/figures#video9

somite expansion and notochord displacement, as well as the *tbx16* loss-of-function NES defect, provides strong evidence that somite morphogenesis is critical for NES.

To further verify the relationship between somite ventral expansion and angioblast migration, we also created a two-variable graph to measure both the relative distance of the notochord to the ventral somite along the dorsal-ventral axis and the percentage of angioblasts arriving at the midline for a given somite. For this analysis, we used 15-somite stage embryos sectioned at the 5th somite and 12th somite (*Figure 7N*, *Figure 7—source data 4*). Embryos sectioned at the 12th somite show little notochord-ventral somite distance and little angioblast migration toward the midline (*Figure 7N*, red squares in gray box). Embryos sectioned at the fifth somite have greater distance from the notochord to ventral somite along the dorsal-ventral axis, as well as significant angioblast migration to the midline (*Figure 7N*, blue dots in orange box, *Figure 7—source data 4*). This corresponds to migration with data seen previously for NES distance, indicating that these processes are tightly correlated.

Finally, we investigated a methodology to track all cellular movements within an embryo to co-visualize somite morphogenesis along with angioblast migration and NES. We used a live explant method to monitor migration in a transverse section view. Live *tg(kdrl:GFP)* embryos, injected with *mCherry-CAAX* mRNA, were sectioned at the 10-somite stage at the fourth somite and mounted to observe a transverse section of the trunk. We were able to observe migrating angioblasts, as well as somite morphogenesis and midline NES in the explant (*Figure 7O and P*, *Video 8*). At the 180′ timepoint, angioblasts were not at the midline and the hypochord remained flush with the endoderm (*Figure 7O*, *Video 8*). At this timepoint, the somites have begun to narrow in the medial-lateral axis. By the 240′ stage, the hypochord is separated away from the endoderm (*Figure 7P*), and shortly after this, the angioblasts are able to complete their migration to the midline by the 270′ timepoint (*Figure 7P*). Similar to the results seen in fixed sections, live explants of embryos treated with DEAB show angioblast migration defects and lack of NES (*Video 9*).

## Discussion

Here, we established a cell-nonautonomous angioblast migration model wherein shape changes in the somitic mesoderm create a physical space for angioblasts to migrate into and develop into the axial vasculature. The emergence of the transient VMC is a necessary requirement for contralateral angioblasts to complete their migration and coalesce at the midline. We observed a defect in somitic mesoderm development in RA-depletion conditions, which delayed somite maturation, intrasomite cellular movement, and significantly impeded NES. Conversely, we found that addition of RA accelerated somite maturation and NES, along with angioblast migration. We were able to confirm that the somitic mesoderm was required for angioblast migration by examining *tbx16* mutants, which have a lack of somites caused by EMT defects (*Goto et al., 2017*; *Ho and Kane, 1990*; *Manning and Kimelman, 2015*; *Row et al., 2011*). Loss of function of *tbx16* prevented angioblast migration completely and resulted in bifurcated blood vessel formation. These changes in vascular morphology indicate that the somitic mesoderm is a requirement for angioblast migration to the midline. It is notable that in amniotes, such as the mouse and quail, the dorsal aorta and cardinal vein are initially bifurcated in a similar manner to zebrafish *tbx16* mutants prior to fusion at the midline (*Drake and Fleming, 2000*; *Pardanaud et al., 1996*; *Pardanaud et al., 1987*). It is possible that evolutionary differences in the somitic mesoderm between teleosts and amniotes could play a role in the differences in vascular development.

Very little is known about the process of NES. Our observations indicate that its initiation correlates very closely with angioblast migration, with angioblasts migrating into the VMC seemingly as soon as it appears in both wild-type and RA-depletion conditions. Given that these blood vessels become lumenized and need to achieve a sufficient volume for blood transport, it is plausible that they require morphological changes within the ventral part of the embryo to accommodate for their size. Delays in somite maturation prevent angioblasts from reaching the midline in a normal developmental time frame. It is possible that the consequences of this migration defect result in the malformed blood vessel structure in RA-depleted embryos, which have small arteries and larger veins. Prior work showed that the most lateral angioblasts, which migrate later than more medial angioblasts, preferentially adopt a venous fate over an arterial one (*Kohli et al., 2013*). This implies that somite maturation is a key factor in determining the proper distribution of endothelial progenitors to the arteries and

veins by essentially timing angioblast migration and altering cell fate decision patterns. However, we cannot definitively determine if this is solely from angioblast migration defects.

We found that the formation of the VMC does not necessarily require the notochord tissue itself. In *noto* morphants, which causes the notochord to adopt a somitic fate, there is still a ventral cavity and angioblasts are able to arrive at the midline. This implies that the notochord is ultimately dispensable for the completion of angioblast migration, even though its loss can produce migration defects (*Helker et al., 2015*; *Sumoy et al., 1997*). The migratory defects could be the result of reduced notochord-secreted factors such as *apela,* which greatly reduce but do not completely abolish migration (*Helker et al., 2015*). Another factor could be that signals from the notochord, such as Hedgehog, influence somite development, which in turn facilitates angioblast migration (*Blagden et al., 1997*; *Hinits et al., 2009*; *Yin et al., 2018*). While the fate of the midline tissue does not need to be notochord for NES to occur, it is likely that mechanical coupling between the somites and midline tissue is required such that somite morphogenesis drives the separation of the midline mesoderm away from the underlying endoderm. In *noto* loss of function, the somitic tissue that forms at the midline in place of notochord is continuous with the flanking somites. In wild-type embryos, there is evidence of mechanical interactions of the somite and notochord during convergent extension, which are required for the remodeling of the adaxial cells, the medial most cells in the somite (*Yin and Solnica-Krezel, 2007*).

The phenotypes observed in both the somites and endothelium after loss of RA signaling are not the result of a simple deficit of mesoderm. RA signaling loss of function affects somite maturation and morphogenesis, but does not prevent them from forming, and yet the migration of the angioblasts is still disrupted. The direct transcriptional targets of RA signaling that induce somite maturation are unknown, but it is clear that RA signaling controls the timing and progress of somite shape changes as they mature. Cell-autonomous effects of RA depletion on the somites, coupled with angioblast migration defects in RA-depleted somites, indicate that RA directly controls somite shape changes that facilitate migration. The fact that somites can adopt mature shapes in explants, even in the absence of adjacent somites, indicates this control is local to individual somites. It is also noteworthy that loss of RA signaling activity in a relatively few number of adaxially localized cells in our chimeric somites was sufficient to cause somite shape and angioblast migration defects. This implies that the somite cells most medial to the midline could be the population required for the ventral expansion of the somite and NES. It is not clear if NES is required for the initiation of angioblast migration as some midline-directed cellular movement occurs in DEAB-treated embryos before NES is visibly seen. It is possible that the somite itself could facilitate the angioblasts' movement to the midline or simply provide an opening through which angioblasts can reach the midline. The fact that angioblasts reside adjacent to the notochord in *tbx16* mutants, but do not migrate beneath them, implies that angioblasts are not sufficient to drive NES or their coalescence at the midline. It is likely that ventral expansions of the somite drive the notochord away from the endoderm. Together, this indicates that somitic maturation requires tight spatial and temporal control to facilitate the concomitant development of the endothelium.

## Materials and methods

**Key resources table**

| Reagent type (species) or resource | Designation | Source or reference | Identifiers | Additional information |
|---|---|---|---|---|
| Antibody | Anti-Etv2 (rabbit polyclonal) | Kerafast | Cat# ES1004; RRID:AB_2904554 | IH (1:500) |
| Antibody | Anti-GFP (mouse monoclonal) | Thermo Fisher | Cat# A11120; RRID:AB_221568 | IH (1:500) |
| Antibody | Secondary antibody, Alexa Fluor 488 (goat polyclonal) | Thermo Fisher | Cat# A11008; RRID:AB_143165 | IH (1:1000) |
| Antibody | Secondary antibody, Alexa Fluor 568 (goat polyclonal) | Thermo Fisher | Cat# A11004; RRID:AB_2534072 | IH (1:1000) |

*Continued on next page*

*Continued*

| Reagent type (species) or resource | Designation | Source or reference | Identifiers | Additional information |
|---|---|---|---|---|
| Antibody | Anti-Digoxigenin-AP, Fab fragments (sheep polyclonal) | Roche | Cat# 11093274910; RRID: AB_514497 | ISH (1:5000) |
| Chemical compound, drug | 4-Nitro blue tetrazolium chloride, solution | Roche | Cat# 11383213001 | |
| Chemical compound, drug | 5-Bromo-4-chloro-3-indolyl phosphate p-toluidine | Roche | Cat# 11383221001 | |
| Chemical compound, drug | DIG RNA Labeling Mix | Roche | Cat# 11277073910 | |
| Peptide, recombinant protein | SP6 RNA polymerase | NEB | Cat# M0207S | |
| Peptide, recombinant protein | T7 RNA polymerase | NEB | Cat# M0251S | |
| Chemical compound, drug | Tricaine-S (MS-222) | Pentair | Cat# TRS1 | |
| Chemical compound, drug | DAPI | Sigma | Cat# D9542 | |
| Chemical compound, drug | 4-Diethylamino benzaldehyde (DEAB) | Sigma | Cat# D86256 | |
| Chemical compound, drug | BMS453 | Cayman Chemical | CAS# 166977-43-1 | |
| Chemical compound, drug | All-trans retinoic acid | Sigma | Cat# R2625 | |
| Other | Tetramethylrhodamine dextran, 10,000 MW, lysine fixable | Invitrogen | Cat# D1817 | |
| Other | Alexa Fluor 647 Dextran, 10,000 MW, Anionic, Fixable | Invitrogen | Cat# D22914 | |
| Other | Modified Barth's Saline (1×), liquid, without Ficoll 400 | Sigma | Cat# 32160801 | |
| Strain, strain background (*Danio rerio*) | Wild-type Tupfel long-fin/AB | N/A | N/A | Wild-type progeny of Tupfel long-fin and AB |
| Strain, strain background (*D. rerio*) | aldh1a2[i26] | *Begemann et al., 2001* | ZFIN ID: ZDB-FISH-150901-19358; RRID:ZFIN_ZDB-ALT-000412-8 | |
| Strain, strain background (*D. rerio*) | aldh1a2[i26], tg(kdrl:eGFP)[s843] | *Begemann et al., 2001; Jin et al., 2005* | RRID:ZFIN_ZDB-ALT-000412-8; RRID: ZFIN_ZDB-GENO-170216-13 | |
| Strain, strain background (*D. rerio*) | tg(hsp70l:id3-2A-NLS-KikGR)[sbu105] | *Row et al., 2018* | ZFIN ID: ZDB-ALT-190306-81; RRID:ZFIN_ZDB-ALT-190306-81 | |
| Strain, strain background (*D. rerio*) | tg(hsp70l:eGFP-dnHsa.RARA)[ci1008] | *Brilli Skvarca et al., 2019* | ZFIN ID: ZDB-TGCONSTRCT-190925-6; RRID:ZFIN_ZDB-ALT-190925-15 | |

*Continued on next page*

*Continued*

| Reagent type (species) or resource | Designation | Source or reference | Identifiers | Additional information |
|---|---|---|---|---|
| Strain, strain background (*D. rerio*) | *tg(kdrl:NLS-eGFP)*[ubs1] | *Blum et al., 2008* | ZFIN ID: ZDB-TGCONSTRCT-081105-1; RRID:ZFIN_ZDB-GENO-081105-1 | |
| Strain, strain background (*D. rerio*) | *tg(kdrl:eGFP)*[s843] | *Jin et al., 2005* | RRID:ZFIN_ZDB-GENO-170216-13 | |
| Strain, strain background (*D. rerio*) | *tg(hsp70l:CAAX-mCherry-2A-NLS-KikGR)*[sbu104] | *Goto et al., 2017* | ZFIN ID: ZDB-ALT-170829-4; RRID:ZFIN_ ZDB-ALT-170829-4 | |
| Strain, strain background (*D. rerio*) | *tg(β-actin:GDBD-RLBD)*[ci1001], *Tg(5XUAS:eGFP)*[nkwasgfp1a] | *Mandal et al., 2013* | ZFIN ID: ZDB-FISH-150901-11429; RRID:ZFIN_ ZDB-GENO-131107-12 | |
| Strain, strain background (*D. rerio*) | *tbx16*[b104], *tg(kdrl:eGFP)*[s843] | *Kimmel et al., 1989*; *Jin et al., 2005* | RRID:ZFIN_ZDB-ALT-980224-16; RRID: ZFIN_ZDB-GENO-170216-13 | |
| Strain, strain background (*D. rerio*) | *tg(actc1b:gfp)*[zf13tg] | *ichi et al., 1997* | ZFIN ID: ZDBTGCONSTRCT-070117-83; RRID:ZFIN_ ZDB-GENO-070830-2 | |
| Strain, strain background (*D. rerio*) | *tg(tbxta:kaedae)*[sbu102] | *Row et al., 2016* | ZFIN ID: ZDB-TGCONSTRCT-160321-5; RRID:ZFIN_ZDB-TGCONSTRCT-160321-5 | |
| Strain, strain background (*D. rerio*) | *tg(hsp70l:lifeact-mScarlet)*[sbu110] | This paper | N/A | Transgenic zebrafish line with heat shock-inducible Lifeact |
| Sequence-based reagent | Morpholino: MO-Noto | *Ouyang et al., 2009* | ZFIN ID: ZDB-MRPHLNO-100514-1 | GGGAATCTGCATGGCGTCTGTTTAG |
| Sequence-based reagent | Morpholino: MO1-tbx16 | *Row et al., 2011* | ZFIN ID: ZDB-MRPHLNO-051107-1 | AGCCTGCATTATTTAGCCTTCTCTA |
| Sequence-based reagent | Morpholino: MO2-tbx16: | *Row et al., 2011* | ZFIN ID: ZDB-MRPHLNO-051107-2 | GATGTCCTCTAAAAGAAAATGTCAG |
| Sequence-based reagent | Riboprobe: *etv2* | *Sumanas et al., 2005* | N/A | Anti-sense riboprobe synthesized using DIG-labeled nucleotides |
| Sequence-based reagent | Riboprobe: *cldn5b* | This paper | N/A | Anti-sense riboprobe synthesized using DIG-labeled nucleotides |
| Sequence-based reagent | Riboprobe: *cxcr4a* | This paper | N/A | Anti-sense riboprobe synthesized using DIG-labeled nucleotides |
| Sequence-based reagent | Riboprobe: *cxcl12a* | This paper | N/A | Anti-sense riboprobe synthesized using DIG-labeled nucleotides |
| Sequence-based reagent | Riboprobe: *dab2* | This paper | N/A | Anti-sense riboprobe synthesized using DIG-labeled nucleotides |
| Sequence-based reagent | Riboprobe: *apela* | This paper | N/A | Anti-sense riboprobe synthesized using DIG-labeled nucleotides |
| Sequence-based reagent | Riboprobe: *aplnra* | This paper | N/A | Anti-sense riboprobe synthesized using DIG-labeled nucleotides |

*Continued*

| Reagent type (species) or resource | Designation | Source or reference | Identifiers | Additional information |
| --- | --- | --- | --- | --- |
| Sequence-based reagent | Riboprobe: *aplnrb* | This paper | N/A | Anti-sense riboprobe synthesized using DIG-labeled nucleotides |
| Sequence-based reagent | Primer: *cldn5b* forward | This paper | N/A | GCAGGCTTGTTTGTTCTGATTC |
| Sequence-based reagent | Primer: *cldn5b* reverse | This paper | N/A | CACAAACAAGTGGGTCGCTG |
| Sequence-based reagent | Primer: *apela* forward | This paper | N/A | CCATCCCTCAGAGGACAGAG |
| Sequence-based reagent | Primer: *apela* reverse | This paper | N/A | CATGTTTGGCAGCAGTAGGA |

| Reagent type (species) or resource | Designation | Source or reference | Identifiers | Additional information |
| --- | --- | --- | --- | --- |
| Sequence-based reagent | Primer: *aplnra* forward | This paper | N/A | ATGGAGCCAACGCCGGAAT |
| Sequence-based reagent | Primer: *aplnra* reverse | This paper | N/A | TCACACTTTGGTGGCCAGC |
| Sequence-based reagent | Primer: *aplnrb* forward | This paper | N/A | ATGAATGCCATGGACAACAT |
| Sequence-based reagent | Primer: *aplnrb* reverse | This paper | N/A | TCACACCTTCGTAGCCAGC |
| Sequence-based reagent | Primer: *cxcr4a* forward | This paper | N/A | CTGAAGGAGCTGGAGAAGTC |
| Sequence-based reagent | Primer: *cxcr4a* reverse | This paper | N/A | GCATGTTCATAGTCCAAGGTG |
| Sequence-based reagent | Primer: *cxcr12a* forward | This paper | N/A | GCGGATCTCTTCTTCACACTGC |
| Sequence-based reagent | Primer: *cxcr12a* reverse | This paper | N/A | TTACACACGCTCTGATCGGTC |
| Sequence-based reagent | Primer: *dab2* forward | This paper | N/A | CTCCTTCATTGCTCGTGATGTC |
| Sequence-based reagent | Primer: *dab2* reverse | This paper | N/A | GCCCTGGTTCAGGTTTCTGG |
| Sequence-based reagent | Primer: *lifeact-mScarlet* forward | This paper | N/A | CAAGCTACTTGTTCTTTTTGCAGGA TCCATGGGCGTGGCCGACTTG |
| Sequence-based reagent | Primer: *lifeact-mScarlet* reverse | This paper | N/A | TTCGTGGCTCCAGAGAATCGATTC ACTTGTACAGCTCGTCCATGC |
| Recombinant DNA reagent | Plasmid: *PCRII-cldn5b* | This paper | N/A | Recombinant vector used for riboprobe synthesis |
| Recombinant DNA reagent | Plasmid: *PCRII-apela* | This paper | N/A | Recombinant vector used for riboprobe synthesis |
| Recombinant DNA reagent | Plasmid: *PCRII-aplnra* | This paper | N/A | Recombinant vector used for riboprobe synthesis |
| Recombinant DNA reagent | Plasmid: *PCRII-aplnrb* | This paper | N/A | Recombinant vector used for riboprobe synthesis |

*Continued*

| Reagent type (species) or resource | Designation | Source or reference | Identifiers | Additional information |
|---|---|---|---|---|
| Recombinant DNA reagent | Plasmid: *PCRII-cxcr4a* | This paper | N/A | Recombinant vector used for riboprobe synthesis |
| Recombinant DNA reagent | Plasmid: *PCRII-cxcr12a* | This paper | N/A | Recombinant vector used for riboprobe synthesis |
| Recombinant DNA reagent | Plasmid: *PCRII-dab2* | This paper | N/A | Recombinant vector used for riboprobe synthesis |
| Recombinant DNA reagent | Plasmid: *hsp70l:lifeact-mscarlet* | This paper | pNJP002 | Plasmid used to generate *tg(hsp70l:lifeact-mscarlet)* |
| Software, algorithm | ImageJ/Fiji | NIH-public domain | https://imagej.nih.gov/ij/download.html; RRID:SCR_003070 | |
| Software, algorithm | Imaris | Bitplane | https://www.bitplane.comhttps://www.bitplane.com; RRID:SCR_007370 | |
| Software, algorithm | Excel | Microsoft | https://www.microsoft.com/en-us/microsoft-365/excel; RRID:SCR_016137 | |
| Software, algorithm | GraphPad Prism 8.4.2 | GraphPad | http://www.graphpad.com/; RRID:SCR_002798 | |

## Resource availability

### Lead contact

Further information and requests for resources and reagents should be directed to and will be fulfilled by the llead contact, Benjamin L. Martin (benjamin.martin@stonybrook.edu).

### Materials availability

The fish line generated in this study is available upon request until availability is made at the Zebrafish International Resource Center.

## Generation of the *tg(hsp70l:lifeact-mScarlet)* transgenic line

This transgenic line was generated using the plasmid pNJP002 (*hsp70l:lifeact-mScarlet*) and *tol2* transgenesis (*Kawakami, 2004*; *Kikuta and Kawakami, 2009*). pNJP002 was generated using PCR amplification paired with Gibson Assembly cloning. A restriction digest was performed on a tol2-hsp70 plasmid (*Row et al., 2016*) using the restriction enzymes BamHI-HF and ClaI (NEB bio labs). PCR amplification of plasmid hCCR4 (*lifeact-mScarlet*) was performed using forward primer CAAGCTAC TTGTTCTTTTTGCAGGATCCATGGGCGTGGCCGACTTG and reverse primer TTCGTGGCTCCA GAGAATCGATTCACTTGTACAGCTCGTCCATGC. Gibson Assembly (NEBbuilder HiFi DNA Assembly, NEB bio labs) was performed on restriction-digested hCCR2 and PCR-amplified *lifeact-mScarlet* to generate plasmid pNJP002. The *tg(hsp70l:lifeact-mScarlet)* [sbu110] line was generated by injecting 25 pg of pNJP002 with 25 pg of *tol2* mRNA using methods described previously (*Row et al., 2016*).

## In situ hybridization and immunohistochemistry

Whole-mount in situ hybridization was performed as previously described (*Griffin et al., 1995*). An antisense RNA probe was synthesized as previously described for *etv2* (*Sumanas et al., 2005*). The *cldn5b, cxcr4a, apela, aplnra, aplnrb, cxcl12a*, and *dab2* DIG probes were generated by *taq* polymerase-generated PCR fragments integrated into a pCRII vector using the Topo-TA Cloning kit (Thermo Fisher). Immunohistochemistry was performed by fixing embryos overnight in 4% paraformaldehyde (Sigma) dissolved in phosphate-buffered saline (PBS) containing 0.1% Tween-20 (PBST) at 4°C. Permeabilization was performed by the placing the embryos in PBS containing 2% Triton X-100 (Sigma) for 1 hr with agitation. Embryos were washed 3× with PBST and blocked in a blocking solution (PBST, 10% goat serum, 1% BSA) for 2 hr at room temperature. Primary antibodies, Etv2 Rabbit

Polyclonal and GFP mouse monoclonal, were diluted to 1:500 in blocking solution and incubated overnight at 4°C. Embryos were washed 5× with PBST for 20 min each wash. Embryos were then incubated in blocking solution for 2 hr, and then incubated in the secondary antibodies Alexa Fluor 488 and Alexa Fluor 568 diluted 1:1000 in blocking solution at 4°C overnight. DAPI stains were done in a 10 µg/mL dilution in PBST for at least 1 hr with agitation.

## Histological analysis

Sections were fixed overnight in 4% paraformaldehyde (Sigma) dissolved in PBS containing at 4°C. Nuclei labeling was done for whole embryos in a 10 µg/mL dilution of DAPI in PBST for at least 1 hr with agitation. Sections, after fixation and DAPI staining, were done in PBST with a 0.15 mm microknife (Fine Science Tools) at the indicated somite location for embryos up to 24 hpf. At 24 hpf or older, embryos were sectioned at approximately the seventh somite. Sections were mounted in 1% agarose in PBS in a 35-mm glass-bottom dish with uncoated #1.5 coverslip and 20 mm glass diameter (MatTek).

## Microinjections

A mix of two *tbx16* morpholinos (MO1: AGCCTGCATTATTTAGCCTTCTCTA [1.5 ng] and MO2: GATG TCCTCTAAAAGAAAATGTCAG [0.75 ng]) were prepared as previously reported and injected into *tg(kdrl:GFP)* embryos (*Row et al., 2016*). 2.5 ng of control morpholino and *noto* morpholino were injected into *tg(kdrl:GFP)* embryos (*Ouyang et al., 2009*). 200 ng of *mCherry*-CAAX mRNA was injected into *tg(kdrl:GFP)* embryos. A 0.2% dilution of Alexa Fluor 647 Dextran (Invitrogen), or a 2% dilution rhodamine dextran (Invitrogen), was used for indicated injections.

## Microscopy and imaging

DIC and fluorescent time-lapse images of wild-type, *noto* morphants, *aldh1a2* mutants, *tbx16* mutants, and drug-treated embryos were performed using a Leica DMI6000B inverted microscope. Wild-type and control embryos were siblings of the morphants and mutants. Live embryos were mounted in 1% low-melt agarose in embryo media containing 1× tricaine (25× stock 0.4 g/L; Pentair,TRS1) in a 35-mm glass-bottom dish with uncoated #1.5 coverslip and 20 mm glass diameter (MatTek). In situ hybridization experiments were imaged using a M165FC microscope (Leica) equipped with an Infinity 3 camera (Lumenera). Embryos were mounted in either horizontal or flat-mounted configuration in 70% glycerol on glass slides.

Fluorescent images of control, *tbx16,* and *noto morphant* (*Figure 3*) cross sections were imaged on a custom-assembled spinning disk confocal microscope consisting of an automated Zeiss frame, a Yokogawa CSU-10 spinning disk, a Ludl stage controlled by a Ludl MAC6000, and an ASI filter turret mated to a Photometrics Prime 95B camera. All other cross sections, including explants, were performed on a custom-assembled spinning disk confocal microscope consisting of a Zeiss Imager A.2 frame, a Borealis-modified Yokogawa CSU-10 spinning disk, ASI 150 µM piezo stage controlled by an MS2000, and ASI filter wheel, and a Hamamatsu ImageEM x2 EMCCD camera (Hamamatsu C9100-23B). These microscopes were controlled with Metamorph microscope control software (V7.10.2.240, Molecular Devices), and laser power levels were set in Vortran's Stradus VersaLase 8 software. Images were processed in Fiji.

## Drug treatments

RA was depleted in embryos using DEAB and BMS453. Stock concentrations of 20 mM DEAB in DMSO were diluted to working concentrations of 20 µM in embryo media. Stock concentrations of 2 mM BMS453 in DMSO were diluted to working concentrations of 2 µM in embryo media. Treatments were done on shield stage embryos unless otherwise noted. All-trans RA was used to activate the RA pathway. Stock concentrations of all-trans RA were diluted from 1 mM stock to 0.1 µM working concentrations in embryo media. All treatments for all-trans RA were performed in tailbud stage.

## Generation of explants for imaging

Transgenic or mutant embryos were grown to the 10-somite stage and then transferred to Modified Barth's Saline (MBS) (Sigma). In this medium, embryos were anesthetized with 1× Tricaine (25× stock 0.4 g/L; Pentair, TRS1) and had their chorions manually removed by forceps. The embryos were then sectioned with a 0.15 mm microknife (Fine Science Tools). Excess yolk was removed with the microknife,

leaving some yolk to prevent injury to the endoderm, and the embryo was then transferred to 35-mm glass-bottom dish (MatTek) with a fire polished pipette. 1.3% low gelling temperature agarose in MBS was heated to 40°C and placed over the embryo in the proper orientation. The agarose was allowed to solidify, and the explant was time-lapsed using fluorescent and DIC microscopy.

## Quantification and statistical analysis

When determining the fluorescence of angioblasts at the midline of the embryo, we utilized the integrated density feature of Fiji software for image analysis. We calculated the corrected total fluorescence of the area at the midline and of the whole embryo using the notochord as a reference for the midline. Corrected total fluorescence for the midline was calculated as (mean gray value of midline × area of midline) – (mean gray value of background × area of midline). Corrected total fluorescence for the embryo was calculated as (mean gray value of embryo × area of embryo) – (mean gray value of background × area of embryo). Percentages for fluorescent intensity were calculated as (corrected total fluorescence at the midline)/(corrected total fluorescence of the whole embryo). Tracks for cells were made using the Trackmate plug-in for Fiji (*Tinevez et al., 2017*). We utilized a Laplacian of Gaussian detector and a Simple LAP Tracker to generate the tracks and analysis. p-Values for indicated figures were generated using two-tailed unpaired Student's *t*-tests. Graphs were generated using GraphPad Prism 8.4.2 for dot plots, and Excel for line graphs. The black lines indicate the median and interquartile ranges, and were generated using the Prism software. Simple linear regression was also determined using GraphPad software.

## Acknowledgements

We thank David Matus for critical review of the manuscript, as well as members of the Martin and Matus labs for helpful comments. We thank Jesus Torres Vazquez for sending the kdrl transgenic reporter lines. We thank Neal Bhattacharji and Stephanie Flanagan for excellent zebrafish care. We thank Tianying Chen for assistance in data acquisition. We also thank Rebecca Adikes, Thom Geer, and Nobska Imaging for microscopy support. The illustration of the 15-somite stage zebrafish embryo was created by Biorender.com. This work was supported by the NSF (IOS 1452928) and NIH NIGMS (R01GM124282, T32GM008468).

## Additional information

### Funding

| Funder | Grant reference number | Author |
|---|---|---|
| National Science Foundation | IOS 1452928 | Benjamin L Martin |
| National Institute of General Medical Sciences | R01GM124282 | Benjamin L Martin |
| National Institute of General Medical Sciences | T32GM008468 | Eric Paulissen |

The funders had no role in study design, data collection and interpretation, or the decision to submit the work for publication.

### Author contributions

Eric Paulissen, Conceptualization, Data curation, Formal analysis, Investigation, Methodology, Validation, Visualization, Writing - original draft, Writing – review and editing; Nicholas J Palmisano, Joshua S Waxman, Resources; Benjamin L Martin, Conceptualization, Formal analysis, Funding acquisition, Investigation, Methodology, Project administration, Supervision, Writing – review and editing

### Author ORCIDs

Eric Paulissen ⓘ http://orcid.org/0000-0002-9118-644X
Joshua S Waxman ⓘ http://orcid.org/0000-0002-8132-487X
Benjamin L Martin ⓘ http://orcid.org/0000-0001-5474-4492

### Ethics

This study was performed in strict accordance with the recommendations in the Guide for the Care and Use of Laboratory Animals of the National Institutes of Health. All of the animals were handled according to approved institutional animal care and use committee (IACUC) protocol (#301537) of Stony Brook University.

### Decision letter and Author response

Decision letter https://doi.org/10.7554/eLife.74821.sa1
Author response https://doi.org/10.7554/eLife.74821.sa2

---

## Additional files

### Supplementary files
• Transparent reporting form

### Data availability
Data generated or analysed during this study are included in the manuscript and supporting files.

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
