## [Editor Report]

This article elegantly combines in vivo imaging and cell transplantation studies in zebrafish embryos to reveal how somite morphogenesis impacts early blood vessel formation. The study provides evidence that retinoic acid-driven somite remodeling generates a midline cavity that is critical to the recruitment and coalescence of angioblast blood vessel precursors at the midline. This work provides important new insights into how angioblast migration and blood vessel formation are coordinated by the timely rearrangement of surrounding tissues during embryonic development.

---

## [Decision Letter]

**Decision letter after peer review:**

[Editors’ note: the authors submitted for reconsideration following the decision after peer review. What follows is the decision letter after the first round of review.]

Thank you for submitting your work entitled "Somite morphogenesis is required for axial blood vessel formation" for consideration by *eLife*. Your article has been reviewed by 3 peer reviewers, one of whom is a member of our Board of Reviewing Editors, and the evaluation has been overseen by a Senior Editor. The reviewers have opted to remain anonymous.

Comments to the Authors:

We are sorry to say that, after consultation with the reviewers, we have decided that your work will not be considered further for publication by *eLife*.

The common feeling of the three reviewers is that the manuscript is potentially interesting, suggesting a previously unappreciated role of somite tissue remodeling in axial blood vessel formation. However, the present version is too preliminary and does not identify the mechanisms by which retinoic acid may drive somite remodeling, cavity opening and angioblast migration. Moreover, the assumption that manipulation of somite morphogenesis specifically impacts axial vessel formation by opening a ventral midline cavity, rather than previously detailed roles for the somite in axial vessel formation, is not adequately supported by the data. The reviewers believe that extensive additional work would be required to meet these criticisms, which is beyond what is possible to complete in a standard manuscript revision period.

We hope you find the full reviews attached below helpful.

*Reviewer #1:*

The work of Paulissen et al. presents a new model for early development of the vertebrate vasculature. Using live imaging microscopy in the zebrafish model, Paulissen and colleagues propose that formation of the first axial blood vessels in zebrafish embryos is dependent upon retinoic acid (RA)-mediated remodeling of surrounding somite tissue, which creates a "ventral midline cavity" that facilitates recruitment of angioblasts to the midline. Whereas others have demonstrated that notochord-derived cues (such as Apela) direct angioblasts to the midline, Paulissen et al. provide evidence that the notochord is overall dispensable for axial vessel formation. Alternatively, they propose that migration of angioblasts to the midline is dependent upon RA-driven somite remodeling, which triggers separation of the notochord and endoderm to create the midline cavity that facilitates angioblast recruitment. In support of this concept, Paulissen et al. use genetic and chemical manipulations of RA signaling to demonstrate that (a) angioblast migration to the midline is dependent on RA, (b) RA regulates somite shape and formation of a ventral midline cavity, and (c) angioblasts populate this cavity as they coalesce at the midline. However, the precise mechanisms by which RA drives somite remodeling remain unclear. More importantly, the work lacks experimental evidence confirming that RA-mediated midline cavity formation is itself is a driver for recruitment of angioblasts. Hence, whilst this study proposes an exciting new model for axial vessel formation, conclusions that RA-mediated somite remodeling drives this process need further clarification. In particular:

1) How does RA drive expansion of a ventral midline cavity?

A number of key questions remain regarding the role of RA in somite remodeling and cavity formation. For example, which RA-responsive tissue drives ventral midline cavity expansion? Indeed, only the notochord, adaxial cells and epidermis appear RA-responsive at these stages (Figure 3) – but not the somite cells that exhibit movement defects in the absence of RA in Figure 7. In Figure 4, transplantation of dnRAR-expressing cells indicates that angioblasts do not require RA signaling for midline migration. Similar experiments defining the role of notochord and/or adaxial or somite tissue in cavity formation would at least shed light on the cell-cell interactions driving cavity formation. For example, can transplanted WT notochord, adaxial or somite tissue recover local angioblast migration in RA signaling deficient mutants. Moreover, such experiment would also be critical to defining the tissue to target in experiments described below.

2) Is formation of a ventral midline cavity required for angioblast migration?

Paulissen et al. provide evidence that formation of a cavity at the midline is dependent on RA signaling, but it still remains unclear if the defects in angioblast migration observed in the absence of RA signaling are directly due to failure of midline cavity formation or are a consequence of disruption to other RA-dependent processes. For example, mosaic experiments using a mix of transplanted WT cells of the relevant cell type (which has yet to be defined – please see point 1 above) in RA-deficient mutant embryos to induce local recovery of cavity formation (or vice versa to disrupt local cavity formation), would determine if angioblasts can only populate regions at the midline where the ventral cavity forms. Without such direct testing of this core hypothesis, evidence that midline cavity formation directly impacts angioblast migration remains only correlative, and strong statements in the text, such as "The emergence of the VMC is a necessary requirement for endothelial progenitors to both migrate into and develop into functioning blood vessels", lack sufficient experimental support.

In addition to experiments suggested above, several other experimental improvements would significantly strengthen the manuscript:

1) Overall, the live imaging data would greatly benefit from higher spatial resolution and robust quantification. For example, live imaging data of notochord-endoderm separation presented in Figure 4 are of poor resolution and are difficult to interpret. These analyses would benefit from post-acquisition conversion to transverse movie sections to directly observe cavity formation and enable direct quantification of cavity size. Likewise, live-imaging of angioblast migration would be strengthened by better spatial resolution and consistent quantification (for example, quantification is absent in Figure 3, 5 and 6), as well as analysis over longer time periods to determine to what extent loss of RA signaling truly blocks angioblast migration, or if this is a slight temporal delay. Importantly, comments in the text such as "Angioblasts in these embryos show no clear anterior to posterior migration patterns and are often disorganized in their directionality" need to be strengthened by appropriate quantification.

2) The author's core hypothesis is that somite morphogenesis creates a midline cavity that is required for recruitment of angioblasts to the midline. Yet, this the study lacks live imaging evidence that this cavity opens prior to angioblast midline coalescence. Experiments in Figure 5 that monitor angioblast migration in tissue slices do not show the midline cavity. As such, further live imaging studies using a membrane label to pinpoint the forming cavity (such as the author's HS:mcherry-caax line) should be performed to provide convincing evidence that cavity formation occurs prior angioblast arrival.

3) Likewise, it would be important to test if angioblasts can still reach the midline in the absence of a midline cavity. For example, angioblast migration defects in aldh1a2 mutants clearly recover later in development, as axial vessels still form (Figure 2). It would be important to determine if the midline cavity needs to be open for this recovery, or if angioblasts can still reach the midline in its absence – which may question the overall importance of cavity formation.

4) As angioblast phenotypes observed upon disruption of RA signaling are near-identical to those described Apela mutants (Helker et al., *eLife*, 2015), it is important to determine that notochord expression of Apela is unaffected by disruption of RA signaling – i.e., that angioblast migration defects are not simply due to loss of notochord Apela expression. This is especially important considering that the notochord is the most RA-responsive tissue at the timepoints investigated (Figure 3).

5) Caution should be exercised when interpreting results in tbx16 mutants that lack somite tissue, as this will perturb expression of many somite-derived cues that impact angioblast behavior. As such, disruption of angioblast migration cannot be solely attributed to a lack of midline cavity formation in tbx16 mutants and text comments, such as "This indicates that the somatic mesoderm is required for NES and formation of the VMC, which in turn is necessary for the angioblasts to reach the midline", need to be toned down to accurately reflect the data.

6) Labelling of Figure 7I, J and K do not match the figure legend and text discussion of the figure.

*Reviewer #2:*

In this article, the authors present a study of midline angioblast migration in zebrafish that requires surrounding tissue rearrangement controlled by somite morphogenesis. This study reveals the role of retinoic acid in the anterior to the posterior progression of midline angioblast migration and how this impacts the opening of the ventral midline cavity. The morphogenetic questions that the authors tackle are interesting ones. The submissions main data set is the description of morphological defects that occur during vascular formation induced by somite maturation and RA signalling. However, this is a study is lacking in resolution and imaging quality to support it assumptions and is somewhat lacking in depth of insight mechanistically into the processes understudy. Furthermore, the somite's influence on the vasculature has, to a degree, been previously detailed. Also, the two separate sections appear to be in essence two different stories that both probably each need more mechanistic insight to be standalone studies.

There are essentially two separate issues dealt with in this submission. The first part of this study proposes the role of retinoic acid in the convergence of angioblast to the midline and the vasculature formation and they suggest retinoic acid induces angioblast migration cell non-autonomously.

The second part of this study proposes the somite maturation as a critical event that is required for Notochord- Endoderm separation and development of the axial vasculature.

General Comments.

The morphogenetic question that the authors tackle is an interesting one. The submissions main data set is the description of morphological defects that occur to vascular formation during somite maturation and RA signalling. However, this is a study is lacking in resolution and imaging quality to support it assumptions and is somewhat lacking in depth of insight mechanistically into the processes understudy. Furthermore, the somite's influence on the vasculature has, to a degree, been previously detailed. Also, the two separate sections appear to be in essence two different stories that both probably each need more mechanistic insight to be standalone studies.

Specific Comments

1) In "Figure 1" the quality of the images are poor and very difficult to discern what is actually happening at cellular resolution.

2) What is the reason to choose the 10 somite stages to study migration?

3) It is mentioned (page 5) DEAB is used at different stages of development, however, just 2 stages (tailbud and 5 somite stage) have shown, are there any changes in other time points of development?

4)It is mentioned Aldh1a2 -/- shows a large reduction in the size of DA and increases the size of the posterior cardinal vein, the results and images are not that convincing (looks like collapse structure), High-resolution sections, Antibody staining with different arterial markers/ qPCR or in situ hybridization in specific genes might support the results.

5) Figure 4, is there any possibility to get the better quality of the images (H-M)?

6)Figure 4: N-P Is there any quantification approach to calculate the size of movement dorsally from the yolk?

7) Figure 6: A/ B/ E/F in black and white is not clear, maybe needs both black and white and color ones beside each other, also needs to be in high resolution with an arrow to mention the area of interest. The title of figure legend: "Genes which affect angioblast migration also affect somitic mesoderm and ventral cavity formation" needs more experiments to prove this as in situ hybridization OR qPCR of various genes.

8. The somites are a source of both endothelial progenitor cells themselves (a fact ignored in this study) as well as signals that regulate them. How are these two observations/properties reconciled in your model. How are they explained in the defects observed?

*Reviewer #3:*

This manuscript, 'Somite morphogenesis is required for axial blood vessel formation' by Paulissen, et al., aims to elucidate novel mechanisms linking somatogenesis to midline migration of angioblasts for axial blood vessel formation. The work is clearly explained and nicely presented, but the studies lack some depth that would increase the impact of this work. Of note, there are previous zebrafish studies/mutants that have been published that demonstrate that abnormal somite formation/cellular delamination leads to mis-patterned vasculature that are not cited or discussed within this current study: (DOI: 10.1002/dvdy.20814, doi.org/10.1101/2020.05.14.096305, DOI:10.1002/dvdy.22410, doi.org/10.1038/s41467-020-16515-y, doi.org/10.1186/1471-213X-10-96, and so on…). Because of this, it is unclear that this work presents the conceptual advancement that it purports, though it is still a well-executed study that adds to this literature. Given the limitations presented above, and the data specific critiques presented below, I cannot recommend publication of this manuscript in its current form.

Concerns:

1. The predominate concern of this reviewer is the concept that the somites are 'required' for axial blood vessel formation. In short, it seems from this work that what is 'required' is space for the cells to move and form tubes, i.e., that the somites move out of the way to allow expansion of cells from the LPM to form the axial vasculature. It was unclear if the authors were suggesting there were additional guidance cues required and produced by the somites, but this did not seem to be the argument. In this case, the conclusions seem very overstated.

2. In several instances data that seems like it would not share a control is plotted on the same graph and represented by the same control data. This should be fixed (examples: Figure 1E, Figure 4G).

3. Figure 2- the authors aim to understand the role of RA signaling in global architecture of the axial vessels. They use an aldh1a2-/- mutant to do so. The initial axial vessel phenotypes are shown at 5dpf, well after these vessels are formed. So, it is impossible to interpret if the small aorta is a primary or secondary defect to loss of aldh1a2. Further, the authors show lumenization at 25hpf, which drastically predates the 5dpf phenotype described. In short, this does not conclusively prove that the DA is still open and not collapsed at 5dpf. The presence/absence of nuclei making up the DA and CV also does not prove that angioblasts preferentially went to the CV versus the DA as the authors also did not assess things such as proliferation to determine if cells are dividing at the same rate in the mutant vs. WT siblings. Without live imaging across a broad timescale and proliferation analysis, this data is over interpreted.

4. The aldh1a2-/- mutants show slower dorsal movement of the notochord than WT siblings. It is unclear why the images in Figure 4H-M are of a chemical treatment instead of the mutants quantified in panel G. Additionally, does exogenous RA rescue the mutant phenotype? This would increase the confidence that the treatments are specific and targeted and strengthen the argument that RA is guiding this dorsal movement phenotype.

5. Do the noto MO treated embryos form both a DA and CV? From the cross sections, it looks as if only one axial vessel is present.

6. Supplemental Movie 4- The transplanted magenta cells don't seem to extend processes like the other cells surrounding them. Are these magenta cells actively or passively moving with the green cells? While they do reach the midline, it's not clear that they are actively 'leading' themselves there. Please clarify, as this is critical for the interpretation of the autonomous/ non-autonomous need for RA signaling in angioblasts. What proportion of these transplanted cells ultimately target to the endothelial lineage? Are there cells in the surrounding tissue that also carry these markers that are not angioblasts?

7. If the tbx16 mutants don't have a functional somatic mesoderm, how and where do the angioblasts arise from? Isn't the LMP derived from the somatic mesoderm? Do the authors know if the same number of angioblasts are present? Are they specified correctly to begin with? The interpretation of this experiment should be clarified as it's unclear if cells that can be termed 'angioblasts' remain present and raises a potential alternate interpretation of this data. Additional studies/clarity needs to be added to address this point.

8. Figure 7I-K: Perhaps these sections aren't representative? It looks like there is separation in J- the DEAB treated. Or are these images mislabeled?

9. Could the authors clarify this point: "Cells in DMSO treated embryos showed significant amounts of displacement throughout the somite (Figure 7C). However, the DEAB cells showed less displacement over time (Figure 7C). Quantification of speed in the same manner showed no statistically significant difference between DMSO and DEAB treatment (Figure 7D)."

How is this possible? If the cells move less distance over time, shouldn't that equate to a change in speed? Additionally, the tracks in E,F are not visible. Please adapt these images.

[Editors’ note: further revisions were suggested prior to acceptance, as described below.]

Thank you for resubmitting your work entitled "Somite morphogenesis is required for axial blood vessel formation" for further consideration by *eLife*. Your revised article has been evaluated by Didier Stainier (Senior Editor) and a Reviewing Editor.

The manuscript has been improved but there are some remaining issues that need to be addressed, as outlined below:

Essential revisions:

Both reviewers agree that an additional transplantation experiment is required to support claims that the ventral midline cavity is itself required for angioblast accumulation at the midline. This experiment requires repetition of transplantation experiments from Figure 7C and D alongside angioblast staining (for Etv2, as in Figure 2B) to allow direct quantification of cavity size and angioblast accumulation at perturbed and unperturbed sites within individual embryos. This would provide stronger evidence that cavity opening itself impacts angioblast accumulation, as well as provide evidence to explain why transplantation of dnRAR-expressing somite tissue disrupts midline accumulation of angioblasts from the right, but not the left side of the embryo in Figure 2H.

*Reviewer #1:*

Following the previous submission of this work for publication in *eLife*, Paulissen et al. now present a significant revision to their study that includes new experimental data, analyses and interpretation. In this re-submission, the authors reveal that retinoic acid (RA) drives angioblast accumulation at the zebrafish embryo midline to form axial blood vessels. Supported by new transplantation and imaging data, they find that RA signaling in the somite acts to drive midline angioblast migration, and that RA drives remodeling of the somite, promotes separation of the notochord and endoderm, and forms a midline cavity. Observations that the opening of this cavity occurs concomitant with arrival of angioblasts leads the authors to propose that progressive somite-mediated opening of this ventral midline cavity is important for midline angioblast accumulation. In response to comments from the reviewers, new data has been included in this re-submission that addresses most of my previous concerns. However, there are still several key points that are not adequately addressed and that should be tackled with further experiments before acceptance for publication.

1) A previous concern was that it is unclear if the ventral midline cavity is itself required for angioblast accumulation at the midline (as proposed by the authors) or, alternatively, that other RA-dependent signals released by the somite are responsible. Observations of concomitant cavity opening and arrival of angioblasts at the midline are suggestive, but not conclusive evidence that cavity opening is required for angioblast accumulation. Repetition of transplantation experiments from Figure 7C and D alongside angioblast staining (for Etv2, as in Figure 2B) would allow direct quantification of cavity size and angioblast accumulation at perturbed and unperturbed sites within individual embryos. Further quantification of cavity size and angioblast accumulation in this experimental condition would provide stronger evidence that local manipulation of ventral midline cavity size has a predictable impact local angioblast migration.

2) New data presented in Figure 2H raise further questions about the role of ventral midline cavity opening in angioblast midline accumulation. In this experiment, transplantation of dnRAR-expressing somite tissue disrupts midline accumulation of angioblasts from the right, but not the left side of the embryo. If the delay to angioblasts accumulation from the right is entirely due to a disruption in ventral midline cavity opening, wouldn't it be assumed that angioblasts access from the left would also be disrupted? The experiments requested in comment 1 would be key to resolve this point upon quantification of the relationship between local cavity size and angioblast accumulation.

3) Separation of the notochord and endoderm are proposed to generate an empty cavity that allows access of angioblasts to the midline. This point needs to be further clarified upon co-staining of transverse sections of Lifeact embryos over time (as in Figure 5) with DAPI, or another nuclear label, to conclusively determine that this cavity is acellular, and where and when it opens up.

*Reviewer #2:*

The authors have addressed the majority of my concerns. The paper has been improved with the inclusion of new data.

---

## [Author Response]

[Editors’ note: the authors resubmitted a revised version of the paper for consideration. What follows is the authors’ response to the first round of review.]

Reviewer #1:The work of Paulissen et al. presents a new model for early development of the vertebrate vasculature. Using live imaging microscopy in the zebrafish model, Paulissen and colleagues propose that formation of the first axial blood vessels in zebrafish embryos is dependent upon retinoic acid (RA)-mediated remodeling of surrounding somite tissue, which creates a "ventral midline cavity" that facilitates recruitment of angioblasts to the midline. Whereas others have demonstrated that notochord-derived cues (such as Apela) direct angioblasts to the midline, Paulissen et al. provide evidence that the notochord is overall dispensable for axial vessel formation. Alternatively, they propose that migration of angioblasts to the midline is dependent upon RA-driven somite remodeling, which triggers separation of the notochord and endoderm to create the midline cavity that facilitates angioblast recruitment. In support of this concept, Paulissen et al. use genetic and chemical manipulations of RA signaling to demonstrate that (a) angioblast migration to the midline is dependent on RA, (b) RA regulates somite shape and formation of a ventral midline cavity, and (c) angioblasts populate this cavity as they coalesce at the midline. However, the precise mechanisms by which RA drives somite remodeling remain unclear. More importantly, the work lacks experimental evidence confirming that RA-mediated midline cavity formation is itself is a driver for recruitment of angioblasts. Hence, whilst this study proposes an exciting new model for axial vessel formation, conclusions that RA-mediated somite remodeling drives this process need further clarification. In particular:1) How does RA drive expansion of a ventral midline cavity?A number of key questions remain regarding the role of RA in somite remodeling and cavity formation. For example, which RA-responsive tissue drives ventral midline cavity expansion? Indeed, only the notochord, adaxial cells and epidermis appear RA-responsive at these stages (Figure 3) – but not the somite cells that exhibit movement defects in the absence of RA in Figure 7. In Figure 4, transplantation of dnRAR-expressing cells indicates that angioblasts do not require RA signaling for midline migration. Similar experiments defining the role of notochord and/or adaxial or somite tissue in cavity formation would at least shed light on the cell-cell interactions driving cavity formation. For example, can transplanted WT notochord, adaxial or somite tissue recover local angioblast migration in RA signaling deficient mutants. Moreover, such experiment would also be critical to defining the tissue to target in experiments described below.

We agree that determining the RA-responsive tissue would strengthen our claims. We performed transplantations of dnRAR expressing cells into the notochord and somites of host angioblast reporter embryos. Transplants of HS:dnRAR cells into the somites showed significant angioblast migration defects, which was not seen in transplants to the angioblast or notochord tissue populations (Figure 2E-2J). This result indicates that the somites are the tissues required for RA mediated midline angioblast migration.

2) Is formation of a ventral midline cavity required for angioblast migration?Paulissen et al. provide evidence that formation of a cavity at the midline is dependent on RA signaling, but it still remains unclear if the defects in angioblast migration observed in the absence of RA signaling are directly due to failure of midline cavity formation or are a consequence of disruption to other RA-dependent processes. For example, mosaic experiments using a mix of transplanted WT cells of the relevant cell type (which has yet to be defined – please see point 1 above) in RA-deficient mutant embryos to induce local recovery of cavity formation (or vice versa to disrupt local cavity formation), would determine if angioblasts can only populate regions at the midline where the ventral cavity forms. Without such direct testing of this core hypothesis, evidence that midline cavity formation directly impacts angioblast migration remains only correlative, and strong statements in the text, such as "The emergence of the VMC is a necessary requirement for endothelial progenitors to both migrate into and develop into functioning blood vessels", lack sufficient experimental support.

We found that VMC formation likely does not initiate angioblast migration to the midline, as we present new data showing some angioblast migration toward the midline can occur before VMC formation (Figure 5G). However, we added additional evidence that midline convergence beneath the notochord cannot occur without NES (Figure 5N, 5O, supplemental movie 8). We updated this view with the following statements “The emergence of the transient VMC is a necessary requirement for contralateral angioblasts to complete their migration and coalesce at the midline.”

In addition to experiments suggested above, several other experimental improvements would significantly strengthen the manuscript:1) Overall, the live imaging data would greatly benefit from higher spatial resolution and robust quantification. For example, live imaging data of notochord-endoderm separation presented in Figure 4 are of poor resolution and are difficult to interpret. These analyses would benefit from post-acquisition conversion to transverse movie sections to directly observe cavity formation and enable direct quantification of cavity size. Likewise, live-imaging of angioblast migration would be strengthened by better spatial resolution and consistent quantification (for example, quantification is absent in Figure 3, 5 and 6), as well as analysis over longer time periods to determine to what extent loss of RA signaling truly blocks angioblast migration, or if this is a slight temporal delay. Importantly, comments in the text such as "Angioblasts in these embryos show no clear anterior to posterior migration patterns and are often disorganized in their directionality" need to be strengthened by appropriate quantification.

For the reviewer’s first point, we address the difficulty in interpreting notochord translocation in the embryo. To better observe the dorsal translocation of the notochord and NES, we utilized a cell-surface marker transgenic line tg(ubb:lck-mNG) to label all cells in the embryo. Using this, we can more accurately visualize notochord translocation (Figure 4A-4C). We saw a similar event, which we were able to quantify, in Figure 4D-4F. To the latter point, we used a new transgenic line, Tg(hsp70l:lifeact-mScarlet), to perform sections at more developmental time points and were able to capture the formation of the VMC (Figure 5A), as well as NES and angioblast migration recovery in RA depletion conditions (Figure 5F-5I). We also were able to perform a novel transverse section explant method to observe angioblast migration to the midline at high resolution, and were able to observe a transient VMC that forms immediately prior to the arrival of angioblasts at the midline and the completion of migration (Supplemental Movie 8).

2) The author's core hypothesis is that somite morphogenesis creates a midline cavity that is required for recruitment of angioblasts to the midline. Yet, this the study lacks live imaging evidence that this cavity opens prior to angioblast midline coalescence. Experiments in Figure 5 that monitor angioblast migration in tissue slices do not show the midline cavity. As such, further live imaging studies using a membrane label to pinpoint the forming cavity (such as the author's HS:mcherry-caax line) should be performed to provide convincing evidence that cavity formation occurs prior angioblast arrival.

We agree that while NES was presumed based on previous data, we had not captured it at high resolution in our manuscript. Using a new transgenic line, Tg(hsp70l:lifeact-mScarlet), we performed sections at more developmental time points and were able to capture the formation of the VMC (Figure 5A). We also were able to perform a novel explant method to observe migrating angioblasts in transverse sections, and were able to determine that a transient VMC forms immediately prior to the completion of migration (Figure 7N, Supplemental Movie 8).

3) Likewise, it would be important to test if angioblasts can still reach the midline in the absence of a midline cavity. For example, angioblast migration defects in aldh1a2 mutants clearly recover later in development, as axial vessels still form (Figure 2). It would be important to determine if the midline cavity needs to be open for this recovery, or if angioblasts can still reach the midline in its absence – which may question the overall importance of cavity formation.

We performed serial sections at later developmental timepoints in DEAB treated embryos and found that the VMC formed later in development, and angioblasts arrived there as it opened (Figure 5H). This indicates that recovery is dependent on VMC formation.

4) As angioblast phenotypes observed upon disruption of RA signaling are near-identical to those described Apela mutants (Helker et al., eLife, 2015), it is important to determine that notochord expression of Apela is unaffected by disruption of RA signaling – i.e., that angioblast migration defects are not simply due to loss of notochord Apela expression. This is especially important considering that the notochord is the most RA-responsive tissue at the timepoints investigated (Figure 3).

We analyzed the Apela/Aplnra/Aplnrb axis via in-situ hybridization and found it is not affected by the loss of RA (Figure Supplement 2).

5) Caution should be exercised when interpreting results in tbx16 mutants that lack somite tissue, as this will perturb expression of many somite-derived cues that impact angioblast behavior. As such, disruption of angioblast migration cannot be solely attributed to a lack of midline cavity formation in tbx16 mutants and text comments, such as "This indicates that the somatic mesoderm is required for NES and formation of the VMC, which in turn is necessary for the angioblasts to reach the midline", need to be toned down to accurately reflect the data.

We clarified out data to better reflect the relative importance of the somitic mesoderm and notochord while avoiding direct comparisons to NES solely from the tbx16 data. The conclusions are changed as follows, “Embryos injected with noto morpholino have angioblasts located at the midline that appear to have resolved into one blood vessel structure (Figure 3H). Embryos injected with tbx16 morpholinos, however, show two distinct vessels located on either side of the notochord (Figure 3I). This indicates that the somites are critical for angioblast convergence at the midline, while the notochord is ultimately dispensable.”

6) Labelling of Figure 7I, J and K do not match the figure legend and text discussion of the figure.

This data was replaced with higher resolution data (Figure 5).

Reviewer #2:[…]Specific Comments1) In "Figure 1" the quality of the images are poor and very difficult to discern what is actually happening at cellular resolution.

While we appreciate this concern, the images in Figure 1 were not meant to show cellular resolution of individual angioblasts, but rather the migration of the collective population of angioblasts. We feel this better illustrates the scope of the angioblast migration defects across the zebrafish trunk. While we improved the clarity of these images, we also now provide images showing defects at the cellular resolution which can now be found in Figure 5 and Supplemental Movies 8 and 9.

2) What is the reason to choose the 10 somite stages to study migration?

Angioblast migration to form the earliest blood vessels initiates at this developmental time point.

3) It is mentioned (page 5) DEAB is used at different stages of development, however, just 2 stages (tailbud and 5 somite stage) have shown, are there any changes in other time points of development?

We chose these stages to determine the critical point in which RA is required for migration. The developmental time period between tailbud and 5-somite stage is small, roughly 2.5 hours of development at standard conditions. We feel that subdividing that time period further would not add significantly to the understanding of the process.

4)It is mentioned Aldh1a2 -/- shows a large reduction in the size of DA and increases the size of the posterior cardinal vein, the results and images are not that convincing (looks like collapse structure), High-resolution sections, Antibody staining with different arterial markers/ qPCR or in situ hybridization in specific genes might support the results.

While it is difficult to say for certain if the smaller DA was the direct result of the angioblast migration defect, this hypothesis is derived from studies that show slower migrating angioblasts preferentially join the cardinal vein (Kohli et al., 2013). We clarified this point in the text, “While we cannot definitively say the migration defects themselves caused the arterial reduction, it indicates that RA is required for the proper formation of the definitive vasculature.”

5) Figure 4, is there any possibility to get the better quality of the images (H-M)?

We replaced these images with more detailed images that better show the midline defect and lack of dorsal notochord movement (Figure 4A-4F).

6)Figure 4: N-P Is there any quantification approach to calculate the size of movement dorsally from the yolk?

We were able to define this data as shown Figure 4G, quantifying how dorsal translocation of the notochord changes in response to RA modulation.

7) Figure 6: A/ B/ E/F in black and white is not clear, maybe needs both black and white and color ones beside each other, also needs to be in high resolution with an arrow to mention the area of interest. The title of figure legend: "Genes which affect angioblast migration also affect somitic mesoderm and ventral cavity formation" needs more experiments to prove this as in situ hybridization OR qPCR of various genes.

We clarified the data presented with the new title, “The somitic mesoderm, not the notochord, is required for midline convergence of angioblasts.” to better reflect the figure contents. We replaced images with clearer images and added arrows to indicate the midline area of interest.

8. The somites are a source of both endothelial progenitor cells themselves (a fact ignored in this study) as well as signals that regulate them. How are these two observations/properties reconciled in your model. How are they explained in the defects observed?

Somite derived endothelial cells can be a source, but only a very small contribution (Ambler et al.,2001). As well, the tbx16 mutants have been shown to preserve endothelial cells in the absence of somitic tissue (Thompson et al., 1998). It has also been established that the lateral plate mesoderm is a prominent source of angioblasts (Prummel et al., 2020).

Reviewer #3:This manuscript, 'Somite morphogenesis is required for axial blood vessel formation' by Paulissen, et al., aims to elucidate novel mechanisms linking somatogenesis to midline migration of angioblasts for axial blood vessel formation. The work is clearly explained and nicely presented, but the studies lack some depth that would increase the impact of this work. Of note, there are previous zebrafish studies/mutants that have been published that demonstrate that abnormal somite formation/cellular delamination leads to mis-patterned vasculature that are not cited or discussed within this current study: (DOI: 10.1002/dvdy.20814, doi.org/10.1101/2020.05.14.096305, DOI:10.1002/dvdy.22410, doi.org/10.1038/s41467-020-16515-y, doi.org/10.1186/1471-213X-10-96, and so on…). Because of this, it is unclear that this work presents the conceptual advancement that it purports, though it is still a well-executed study that adds to this literature. Given the limitations presented above, and the data specific critiques presented below, I cannot recommend publication of this manuscript in its current form.

We understand that somites have been shown to have some effect on the definitive vasculature. However, none have focused on midline angioblast migration, nor the role of somite morphogenesis, specifically, on the formation of the midline vasculature. We reflect on this with the following text, “While some evidence has shown somite-related defects on blood vessel patterning, much of this research focused on angiogenic sprouting or defects in arterial-venous specification, well after angioblast migration and somitogenesis is completed (Lawson et al., 2002; Shaw et al., 2006; Therapontos and Vargesson, 2010; Torres-Vázquez et al., 2004).

Concerns:1. The predominate concern of this reviewer is the concept that the somites are 'required' for axial blood vessel formation. In short, it seems from this work that what is 'required' is space for the cells to move and form tubes, i.e., that the somites move out of the way to allow expansion of cells from the LPM to form the axial vasculature. It was unclear if the authors were suggesting there were additional guidance cues required and produced by the somites, but this did not seem to be the argument. In this case, the conclusions seem very overstated.

We now show in Figure Supplement 2 that known angioblast migration cues are unaffected. Additional evidence shows that Notochord translocation (NES) and formation of the VMC are required for the completion of angioblast migration (Figure 4 and 5). Data in figures 3 and 7 also show that lack of somites, or lack of proper morphogenesis of somites (caused by loss of RA signaling), causes a failure of VMC formation. Together the data support a model in which the morphogenesis of the somites induces the VMC, which is in turn required for the angioblasts to reach the midline.

2. In several instances data that seems like it would not share a control is plotted on the same graph and represented by the same control data. This should be fixed (examples: Figure 1E, Figure 4G).

We have separated the graphs and added the proper controls to each graph (Figure 1E, and 1F). We clarified our controls in Figure 4G to demonstrate that all embryos were imaged in the same conditions.

3. Figure 2- the authors aim to understand the role of RA signaling in global architecture of the axial vessels. They use an aldh1a2-/- mutant to do so. The initial axial vessel phenotypes are shown at 5dpf, well after these vessels are formed. So, it is impossible to interpret if the small aorta is a primary or secondary defect to loss of aldh1a2. Further, the authors show lumenization at 25hpf, which drastically predates the 5dpf phenotype described. In short, this does not conclusively prove that the DA is still open and not collapsed at 5dpf. The presence/absence of nuclei making up the DA and CV also does not prove that angioblasts preferentially went to the CV versus the DA as the authors also did not assess things such as proliferation to determine if cells are dividing at the same rate in the mutant vs. WT siblings. Without live imaging across a broad timescale and proliferation analysis, this data is over interpreted.

As indicated above, we agree that it is difficult to say for certain if the smaller DA was the direct result of the angioblast migration defect, although that hypothesis is derived from studies that show slower migrating angioblasts preferentially join the cardinal vein (Kohli et al., 2013). We clarified this point in the text, “While we cannot definitively say the migration defects themselves caused the arterial reduction, it indicates that RA is required for the proper formation of the definitive vasculature.”

4. The aldh1a2-/- mutants show slower dorsal movement of the notochord than WT siblings. It is unclear why the images in Figure 4H-M are of a chemical treatment instead of the mutants quantified in panel G. Additionally, does exogenous RA rescue the mutant phenotype? This would increase the confidence that the treatments are specific and targeted and strengthen the argument that RA is guiding this dorsal movement phenotype.

The data quantified in Figure 4G were different images than shown in 4H-4M. We have provided images of that data to illustrate how the Figure 4G data was quantified (Figure 4D-4F). In response to the second point, the aldh1a2-/- mutant targets the dehydrogenase directly responsible for RA production, and we confirmed the specificity of the angioblast migration defect by observing the same phenotype in three additional methods of inhibiting RA signaling (2 pharmacological and one genetic, Figure S1). Additionally, exogenous RA is shown to cause the opposite phenotype of loss of RA (Figure 4C and 4F).

5. Do the noto MO treated embryos form both a DA and CV? From the cross sections, it looks as if only one axial vessel is present.

It was shown previously that noto loss of function results in a single, disorganized blood vessel. This is reflected in the new statement “In noto mutants, it was observed that some angioblasts had reached the midline at later developmental stages, and resolve into a single blood vessel (Fouquet et al., 1997).”

6. Supplemental Movie 4- The transplanted magenta cells don't seem to extend processes like the other cells surrounding them. Are these magenta cells actively or passively moving with the green cells? While they do reach the midline, it's not clear that they are actively 'leading' themselves there. Please clarify, as this is critical for the interpretation of the autonomous/ non-autonomous need for RA signaling in angioblasts. What proportion of these transplanted cells ultimately target to the endothelial lineage? Are there cells in the surrounding tissue that also carry these markers that are not angioblasts?

The cells transplanted in supplemental movie 4 are entirely targeted toward the angioblasts. We utilized a technique to drive cells into the angioblast progenitor pool by excluding them from the somitic mesoderm (Row et al., 2018). Based on new data in which we have targeted dnRAR cells to the somites, which causes the angioblast migration defect, we feel that we have provided strong evidence that RA signaling in somitic mesoderm is affecting angioblast migration rather than cell autonomous effects in angioblasts (Figure 2H).

7. If the tbx16 mutants don't have a functional somatic mesoderm, how and where do the angioblasts arise from? Isn't the LMP derived from the somatic mesoderm? Do the authors know if the same number of angioblasts are present? Are they specified correctly to begin with? The interpretation of this experiment should be clarified as it's unclear if cells that can be termed 'angioblasts' remain present and raises a potential alternate interpretation of this data. Additional studies/clarity needs to be added to address this point.

It has been shown previously that tbx16 mutants have endothelial progenitors (Thompson et al., 1998). Additionally, it has been established that the lateral plate mesoderm is an abundant source of angioblasts (Prummel et al., 2020).

8. Figure 7I-K: Perhaps these sections aren't representative? It looks like there is separation in J- the DEAB treated. Or are these images mislabeled?

This data has been removed in lieu of higher resolution data (Figure 5).

9. Could the authors clarify this point: "Cells in DMSO treated embryos showed significant amounts of displacement throughout the somite (Figure 7C). However, the DEAB cells showed less displacement over time (Figure 7C). Quantification of speed in the same manner showed no statistically significant difference between DMSO and DEAB treatment (Figure 7D)."How is this possible? If the cells move less distance over time, shouldn't that equate to a change in speed? Additionally, the tracks in E,F are not visible. Please adapt these images.

We have added new images with better image tracks (Figure 7I-J). We note that displacement is not the equivalent to total distance travelled, as directionless movement leads many cells to end in roughly the same position as they begin.

[Editors’ note: what follows is the authors’ response to the second round of review.]

Essential revisions:Both reviewers agree that an additional transplantation experiment is required to support claims that the ventral midline cavity is itself required for angioblast accumulation at the midline. This experiment requires repetition of transplantation experiments from Figure 7C and D alongside angioblast staining (for Etv2, as in Figure 2B) to allow direct quantification of cavity size and angioblast accumulation at perturbed and unperturbed sites within individual embryos. This would provide stronger evidence that cavity opening itself impacts angioblast accumulation, as well as provide evidence to explain why transplantation of dnRAR-expressing somite tissue disrupts midline accumulation of angioblasts from the right, but not the left side of the embryo in Figure 2H.

This experiment was performed as suggested using either control wild-type donor cells or dnRAR expressing donor cells transplanted into host embryos with either a ubiquitous mNeonGreen plasma membrane transgene, or host embryos with ubiquitous Lifeact-mScarlet + kdrl:GFP angioblast reporter transgenes. We selected embryos where transplanted cells were localized to somites on one side of the embryo and not the other, to observe a direct comparison between somites on either side in transverse sections. In both types of host embryos, the ventral midline cavity does not open on the side of the embryo where dnRAR cells are located within the somite, but does begin to open on the side where donor cells are absent from the somite. The kdrl:GFP fluorescence indicates this partial opening on the side without transplanted cells facilitates angioblast migration to the midline, whereas angioblasts remain in a more lateral position on the side with dnRAR cells. Control experiments using wild-type donor cells showed no difference in ventral midline cavity opening or angioblast migration on the side with transplanted cells and the side without. These new results are presented in Figure 7 —figure supplement 1.

Reviewer #1:Separation of the notochord and endoderm are proposed to generate an empty cavity that allows access of angioblasts to the midline. This point needs to be further clarified upon co-staining of transverse sections of Lifeact embryos over time (as in Figure 5) with DAPI, or another nuclear label, to conclusively determine that this cavity is acellular, and where and when it opens up.

To show that the ventral midline cavity is acellular, we examined a transverse section image Z-stack of an embryo containing both the ubiquitous Lifeact-mScarlet and the angioblast reporter kdrl:GFP transgenes, and stained with DAPI. A maximum projection of a 15 μm image stack at a time-point immediately preceding angioblast arrival at the midline shows that the ventral midline cavity is devoid of Lifeact and DAPI fluorescence, indicating that it is an acellular cavity. This new result is presented in Figure 5 —figure supplement 1.